

# Satellite observations of gravity wave momentum flux in the mesosphere / lower thermosphere (MLT): feasibility and requirements

Qiuyu Chen[1], Konstantin Ntokas[1], Björn Linder[2], Lukas Krasauskas[1], Manfred Ern[1], Peter Preusse[1], Jörn Ungermann[1], Erich Becker[3], Martin Kaufmann[1], and Martin Riese[1]

[1]Institute of Energy and Climate Research (IEK-7: Stratosphere), Forschungszentrum Jülich, Jülich, Germany
[2]Department of Meteorology, Stockholm University, Stockholm, Sweden
[3]Northwest Research Associates Inc., Boulder, CO, USA

**Correspondence:** Peter Preusse (p.preusse@fz-juelich.de)

**Abstract.** In the recent decade it became evident that we need to revise our picture of how gravity waves (GWs) reach the mesosphere and lower thermosphere (MLT). This has consequences for not just the properties of the GWs itself, but in particular for the global circulation in the MLT. Information on spectral distribution, direction and zonal mean GW momentum flux is required to test the theoretical and modeling findings. In this study, we propose a constellation of two CubeSats for observing

mesoscale GWs in the MLT region by means of temperature limb sounding in order to derive such constraints. Each CubeSat deploys a highly miniaturized spatial heterodyne interferometer (SHI) for the measurement of global oxygen atmospheric band emissions. From these emissions, the 3-D temperature structure can be inferred. We propose to obtain four independent observation tracks by splitting the interferograms in the center and thus gaining 2 observation tracks for each satellite. We present a feasibility study of this concept based on self-consistent, high-resolution global model data. This yields a full chain

of end-to-end (E2E) simulation incorporating 1) orbit simulation; 2) airglow forward modelling; 3) tomographic temperature retrieval; 4) 3-D wave analysis; and 5) GW momentum flux (GWMF) calculation. The simulation performance is evaluated by comparing the retrieved zonal-mean GWMF with that computed directly from the model wind data. A major question to be considered in our assessment is the minimum number of tracks required for the derivation of 3D GW parameters with sufficient accuracy. In particular, our simulations show that the GW polarization relations are still valid in the MLT region and can thus be

employed for inferring GWMF from the 3–D temperature distributions. Based on the E2E simulations for gaining zonal-mean climatologies of GW momentum flux, we demonstrate that our approach is robust and stable, given a four-track observation geometry and the expected instrument noise under nominal operation conditions. Using phase-speed-direction spectra we show also that the properties of individual wave events are recovered when employing four tracks. Finally, we discuss the potential of the proposed observations to address current topics in the GW research. We outline for which investigations ancillary data

are required to answer science questions.



# 1 Introduction

The integration of parametrized gravity waves (GWs) into general circulation models was a tremenduous breakthrough in understanding the mesosphere and lower thermosphere (MLT) region. As they replaced Rayleigh friction, the wave driven circulation and the cold mesopause could be understood (Holton et al., 1995; McIntyre, 1999; McLandress, 1998). However, in their classical formulation, GW parametrizations assume only orography and (often unspecified) non-orographic sources in the troposphere and simplify propagation to be only vertical and instantaneous. In this framework, the waves propagate until they reach either saturation or a critical level and then transfer the dissipated momentum entirely to the background flow. Such interpretations are supported, for instance, by airglow observations and their match with lower level filtering as described by, for instance, blocking diagrams (Taylor et al., 1993). In the last two decades, however, it became evident that this view is too simplified and that the simplifications have important consequences for the large scale dynamics. In order to illustrate this, let us consider three prominent examples where new concepts are essential: (1) the wind reversal above the summer MLT; (2) the recovery phase of sudden stratospheric warmings; and (3) gravity waves in the thermosphere.

**1)** At summer mid latitudes tropospheric winds are westerly, but stratospheric winds are easterly. According to the classical picture this should filter out all GWs with phase speeds up to several tens of m/s in both eastward and westward propagation directions. However, the wind reversal towards westerly winds in the MLT is caused by the dissipation of eastward propagating GWs. How can these then reach the MLT? One conceivable process would be GWs of extremely high phase speeds (on the order of 90 m/s) and very small amplitudes, which would not be visible in the stratosphere but would gain saturation amplitudes at high altitude. This occurs in all GW parametrizations with a wide range of phase speeds (e.g. Alexander and Dunkerton, 1999) and has also been suggested by ray-tracing simulations (Preusse et al., 2009a). A second possibility is lateral GW propagation. Indication for lateral propagation of GWs from subtropical convective regions was first found by Jiang et al. (2004) in Microwave Limb Sounder (MLS) observations of the stratosphere. Observations from the Sounding of the Atmosphere using Broadband Emission Radiometry (SABER) satellite allow to follow the propagation of GWs from 20° latitude at 30 km to 50° latitude at 80 km (cf. Figure 5 of Chen et al. (2019)). Convective GWs therefore remain at all altitudes in an easterly or low wind velocity flow and circumvent the critical levels. Gravity-wave-allowing high-resolution GCM simulations (Liu et al., 2014) are consistent with these observations, and the oblique propagation may be favored by large horizontal wind gradients (Thurairajah et al., 2020). The third alternative is secondary (and higher order) gravity waves. When the original gravity waves from the troposphere break, they exert a body force that excites new gravity waves (Vadas and Fritts, 2002). The relevance of secondary wave generation for the summer MLT is demonstrated by high-resolution GCM simulations (e.g. Becker and Vadas, 2020). There are hence three competing pathways for GWs to reach the summer MLT which need to be distinguished. The way GWs reach the summer MLT necessarily impacts on the phase-speed and direction distribution: only very fast waves in the first case, a noted poleward preference in the second case and waves from breaking regions in the third case. It is likely that all pathways occur simultaneously, but the interaction is complex and not well understood (Thurairajah et al., 2020).

**2)** Gravity waves are believed to play an important role in sudden stratospheric warmings (SSWs) (Thurairajah et al., 2014; Ern et al., 2016; Thurairajah and Cullens, 2022). An SSW event is marked by a general breakdown of the polar vortex and





a major event defined by the wind reversal at 10 hPa. This leads to completely changed propagation conditions for GWs. A new vortex and an elevated stratopause then form at MLT heights and propagate downward. This re-formation at high altitudes is believed to be mainly caused by GWs. Again the question is how the GWs reach the MLT (Thurairajah et al., 2014; Ern et al., 2016). Here also lateral GW propoagation and excitation of GWs in the stratosphere, e.g., by an unstable polar vortex, play a role. Secondary GWs will form at critical levels. For SSWs these additional wave sources not captured by classical GW
parametrisation schemes jumble the large scale dynamics in the MLT. In models with parametrized GWs a strong easterly wind bias forms in the MLT after an SSW, which takes the form of a spourious anticyclone (Harvey et al., 2022). This bias sets in about two weeks after the central date and lasts for more than a month. These results underpin that secondary GWs and other middle atmosphere sources are essential for the resiudal circulation and the related dynamical structure in the winter upper mesosphere (Becker and Vadas, 2018; Stober et al., 2021).

**3** The situation becomes increasingly complex in the thermosphere. Wind reversals below the considered altitude are now almost ubiquitous. Still, observations indicate spatial patterns in the global distributions which correlate to those in the stratosphere (Trinh et al., 2018). This is evidence that at least a larger part of the GWs in the thermosphere are secondary GWs, which preserve the spatial patterns of the primary waves. In addition, waves from other middle atmosphere sources play a role. Understanding the wave sources at the lower boundary of the thermosphere is essential for the whole-atmosphere system,
since GWs vertically couple the thermosphere to the lower atmosphere (e.g., Miyoshi et al., 2014; Park et al., 2014; Yigit and Medvedev, 2015; Yiğit et al., 2016; Vadas and Becker, 2019; Becker and Vadas, 2020). Primary and higher-order GWs reaching altitudes above about 250 km can lead to disturbances or irregularities in the ionospheric layer and thereby affect space-based applications (e.g. Hines, 1960; Bertin et al., 1975; Vadas and Fritts, 2006; Vadas, 2007; Krall et al., 2013; Yiğit et al., 2016; Liu, 2016). In short, together with tides and planetary waves, GWs are the most important dynamical process in
the MLT region. Understanding the various aspects regarding GW instability and transition turbulence, interactions with the ambient large-scale flow, and the generation of higher-order GWs requires extensive knowledge of the spectral and spatial distributions of GWs, including geographical and seasonal variations.

Of course one can tune GW parametrizations in such a way to mitigate some of the observed discrepancies, but overtuning a conceptually wrong parametrization would easily lead to contradictions. For instance, the dissipating GWs generating the wind
reversal in the summer MLT also induce the meridional circulation and hence cause the cold summer mesopause, but for the latter not only GWMF but also energy dissipation and turbulence are of relevance. To simultaneously capture the circulation, the cold summer mesopause and the winter MLT winds in both SSW and non-SSW years therefore requires to represent also the spectral distribution of GWs correctly. This leads us to some higher-level science questions, which to answer is essential for understanding the MLT and the coupling between the middle atmosphere and the thermosphere:

**Science Questions:**

- How do GWs reach the summer MLT?

- Which GWs lead to the formation of the elevated stratopause and the new vortex after an SSW?

- Is there strong westward GW drag in the MLT in the period 2 weeks to 2 months after an SSW?





- – Which GWs propagate into the MLT?

In order to answer these questions we need to characterize the GWs in the MLT region. Key quantities to be determined are:

**Key Quantities**

- – Zonal mean GW momentum flux and its vertical gradient (GW drag)

- – Phase-speed and direction distributions of GWMF

The first key quantity clearly calls for global observations. The second, can in principle be inferred from individual obser-
vations. Ground-based networks, however, will induce biases, as they observe only in the vicinity of land and not over open
oceans.

In general, no observation technique can characterize the entire spectrum of GWs and different kinds of observations need
to be combined for a consistent picture of GWs and their impact in the MLT. For a limited number of locations, spectral
information and GW momentum flux can be inferred from ground-based radar and lidar systems (e.g. Stober et al., 2013; de Wit
et al., 2014; Placke et al., 2015; Bossert et al., 2015, 2018). In addition, ground-based airglow imagers provide information
about GWs with long vertical and short horizontal wavelengths (e.g. Tang et al., 1966; Espy et al., 2006). When it comes to
the large-scale momentum budget in the MLT, these observations are, however, biased. They are made only on land, often in
locations of specific geophysical interest (for example, strong activity of mountain waves). Furthermore, optical systems can
work under clear-sky conditions only.

Based on existing space-borne limb-scanning observations that allowed distributions of the absolute GW momentum flux to
be inferred (e.g. Ern et al., 2004; Alexander et al., 2008a; Preusse et al., 2009a; Ern et al., 2011; Alexander, 2015; Ern et al.,
2018), proposals were made on how a limb-imaging satellite mission could drastically improve our knowledge about GWs in
the stratosphere (Preusse et al., 2009b, 2014). Such an instrument would provide 3D data at good spatial resolution by high
along-track sampling, tomographic retrieval in along-track slices (Ungermann et al., 2010a, b; Song et al., 2017), and across-
track coverage by multiple tracks (illustrated in Fig. 1). For first existing global observations, nadir measurements using the
Atmospheric InfraRed Sounder (AIRS) (Ern et al., 2017; Hindley et al., 2020) were exploited for 3D data. Although these data
capture only long vertical wavelengths, i.e. high intrinsic phase speeds, they provide information on direction characteristics
and allow to demonstrate how backward ray-tracing can be used for source identification from global data (Perrett et al., 2021).
These examples are for the stratosphere only. Nevertheless, it is evident from such studies that a limb imager would provide
novel information about GWs in the MLT.

The spatial sampling drives the complexity of the instrument and hence drives the cost of the mission. The spatial sam-
pling requirements, therefore, need to be justified. The across-track dimension is provided by observing multiple tracks. An
important question therefore is how many parallel tracks are required to gain reliable information about medium-scale GWs
with horizontal wavelengths longer than 100 km, and how these tracks should be spaced. On the one hand, it is obvious that
the wider the overall swath is and the smaller the individual pixels are, the higher the likelihood is to acquire unprecedented
scientific data. On the other hand, a larger number of tracks is a driver for increased instrument complexity and data downlink
rate. The MATS (Mesospheric Airglow/Aerosol Tomography and Spectroscopy) mission (Gumbel et al., 2020), for instance,


is designed to have a $200\,km$ swath width and $5\,km$ wide pixels. MATS is, however, a small-satellite mission with a cost on the order of several tens of millions of Euro. In addition, spatial resolution is achieved at the cost of having only four spectral
sampling points of airglow. An alternative option would be a CubeSat mission which takes fewer tracks of measurements but is still capable of providing a similar amount of information about GWs. Employing a spatial heterodyne spectrometer (SHS), the CubeSat instrument will be better suited to infer spectral information, which means that it will provide a higher resolution of GW spectra. Would one be able to achieve similar aims with a CubeSat mission at a substantially lower cost? One of the aims of this paper is to examine the minimum number of tracks required for deriving 3D wave vectors of GWs from tomographic
temperature observations.

    Airglow emissions at $762\,nm$ from the oxygen atmospheric band ($O_2$ A-band) are particularly suited to gain information on MLT dynamics. Limb observations facilitate high vertical resolution during both day and night. Assuming rotational local thermodynamic equilibrium, the kinetic temperature around the tangent points can then be inferred from the relative line intensities. Using advances in CubeSat standard components, detector technology and optics, a highly miniaturized spatial
heterodyne interferometer (SHI) (Kaufmann et al., 2018) was developed for this purpose. This detection technology can be applied in a CubeSat constellation mission, which consists of two CubeSats, each hosting an SHI. By flying the two SHIs in parallel (illustrated in Fig. 1), and by separately inverting the left-hand and right-hand part of one interferogram, thus splitting the horizontal field of view (FOV) in two, in total four independent observation tracks can be obtained from the proposed satellite observation geometry. High along-track resolution will be achieved using tomographic retrievals.

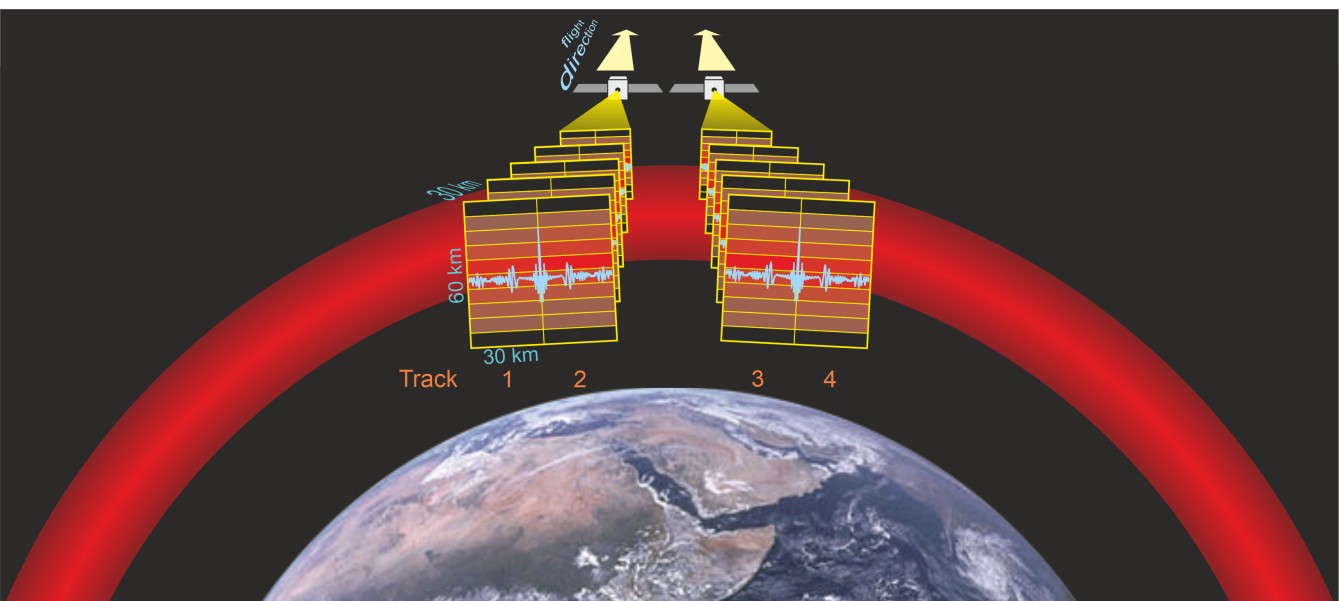

**Figure 1.** Proposed observation geometry of two CubeSats flying in parallel and viewing backward. Each CubeSat carries an SHI which images the atmosphere in the vertical and generates a combined spectral/spatial view in the horizontal. By splitting the interferogram at zero optical path difference, both sides can be evaluated individually, allowing for four effective observation tracks.





Based on limb-sounding four tracks simultaneously, we aim to observe medium-scale GWs of horizontal wavelengths longer than $100\,\mathrm{km}$. In order to demonstrate the feasibility of quantifying GW properties by airglow limb observations, we will perform full chain of end-to-end (E2E) simulations based on self-consistent model data. This validation of our methodology is based on GW parameters (e.g. GW momentum flux) that are derived from temperature residuals using the GW polarization relations (Fritts and Alexander, 2003; Ern et al., 2004; Preusse et al., 2009b). The feasibility will then be evaluated by inferring

such GW parameters directly from the model data. This means that we choose the inferred GW parameters as a performance measure instead of a separate consideration of noise and resolution[1]. In order to study the viability of a CubeSat mission, we will consider the following questions in this study:

**Study Aims:**

– Are polarization relations valid in the MLT region?

    – How few measurement tracks are required? Are 4 or even 2 tracks sufficient?

    – How much instrument noise can we afford in the temperature retrieval and wave analysis?

We address these three questions as follows: The assessment strategy is outlined in Sect. 2. Detailed introductions to models, tools and the instrument are presented in Sect. 3. The outcomes of the assessment and answers to the questions are given in

Sect. 4, followed by a discussion on scientific applications in Sect. 5. Finally, we summarize our findings (Sect. 6).

## 2 Assessment strategy

In this section we outline the strategy to address the questions formulated in the introduction. For our study, we base on fields of winds and temperature from a free-running general circulation model (GCM) HIAMCM (Sect. 3.1) resolving a larger part of the GW spectrum. Since the GCM simulates all dynamical features from first principles, wind and temperature structures

of the model fields are consistent for all waves resolved. To retain this consistency, winds and temperatures are separated into global scale dynamics and GW fluctuations in the same way on the full model fields (Sect. 2.1). For an assessment we need a reference to evaluate the performance against. This is the zonal mean of zonal momentum flux derived from the model winds (Sect. 2.2). With this given, we can design the method (Sect. 2.3) to tackle the three questions posed in the introduction.

### 2.1 Scale separation

The assessment is based on consistent GW-related fluctuations of temperature and wind velocities. This requires a scale separation between GWs on the one hand and global scale dynamics on the other hand. For satellite data, traditionally waves up to zonal wavenumber 6 have been treated as global scale waves and all remaining fluctuations as GWs (e.g. Fetzer and Gille,

---

[1]The alternative approach is to infer typical amplitudes in the MLT and request retrieval noise to be lower than a fraction of this noise. However, there is also noise suppression by regularization and by the spectral analysis software which is difficult to estimate in a forward way in its effects on the visibility function and on the retrieved GWMF distributions





1994; Preusse et al., 2002; Ern et al., 2018). This separation is used, as wavenumber 6 is the highest wavenumber which can be reliably resolved by a single-observation-track low Earth orbit (LEO) satellite (Salby, 1982). For completeness, we have 170 described such a scale separation approach in the Appx. A.

For model studies a much higher separation wavenumber has often been used. Strube et al. (2020) have shown that separation wavenumbers 6 to 8 are sufficient for the stratosphere and that removing wavenumbers up to 40 significantly cuts into the GW part of the spectrum. For studies including the UTLS Strube et al. (2020) recommend zonal wavenumber up to 18 for scale separation. In this study we also use zonal wavenumber 18 with an additional meridional Savitzky-Golay filter of 3rd order 175 polynomials over $5°$ of latitude. This defines our large-scale background which is subtracted from individual temperature values in order to define residuals.

A satellite in a sun-synchronous orbit acquires data at a continuously evolving observation time, but fixed local time for a given latitude on ascending and descending orbit legs. Such a sampling cannot be generated from model data which are sampled at fixed UTC and sampling intervals of O(1 h). Switching between model fields as the orbit evolves would result in 180 jumps at the switching points, while interpolation in time would smooth the interpolated GW fields in an unpredictable manner. In this study we therefore use a single model snap shot (01-Jan-2016 06 UT). Synthetic orbit data generated in this manner allow to address all questions stated in the introduction. In particular, a fixed UTC has the advantage that the synthetic orbit data can also be compared to the reference of zonal mean GWMF from full model fields, which is the basis of our assessment. However, the detrending method via space-time spectral analysis described in Sect. A used for e.g. SABER data (Ern et al., 185 2018) cannot be simulated in this way and the assessment of the background removal as a processing step remains beyond the aims of the current study.

## 2.2 Zonal mean momentum flux: a true reference

At the end, we aim at quantifying the vertical flux of horizontal pseudomomentum of GWs (Fritts and Alexander, 2003)

$$(F_{px}, F_{py}) = \rho \left(1 - \frac{f^2}{\hat{\omega}^2}\right) \left(\overline{u'w'}, \overline{v'w'}\right), \tag{1}$$

where $\rho$ is the background density, $f$ the Coriolis parameter, $\hat{\omega}$ the intrinsic frequency and $u', v', w'$ the wind perturbations due to the GW. The overline denotes the average over a full or multiple wavelengths of the wave. The vertical gradient of the pseudomomentum flux (PGWMF) determines the acceleration of the background wind on a rotating sphere. However, determining $\hat{\omega}$ from a given 3D data set involves some kind of wave analysis and, accordingly, assumptions. On the other hand, the zonal mean of zonal momentum flux ($\rho \overline{u'w'}$) is a true reference as it depends on the wind fluctuations only and as on the 195 cyclical domain of a longitude all waves are properly averaged. If not explicitly stated otherwise, we will hence consider GW momentum flux (GWMF) without the correction for Coriolis force in our assessment. Considering all GWs with horizontal wavelengths longer than 100 km, a regional average or zonal mean of pseudomomentum flux is about 20-30% lower than the same average for momentum flux.





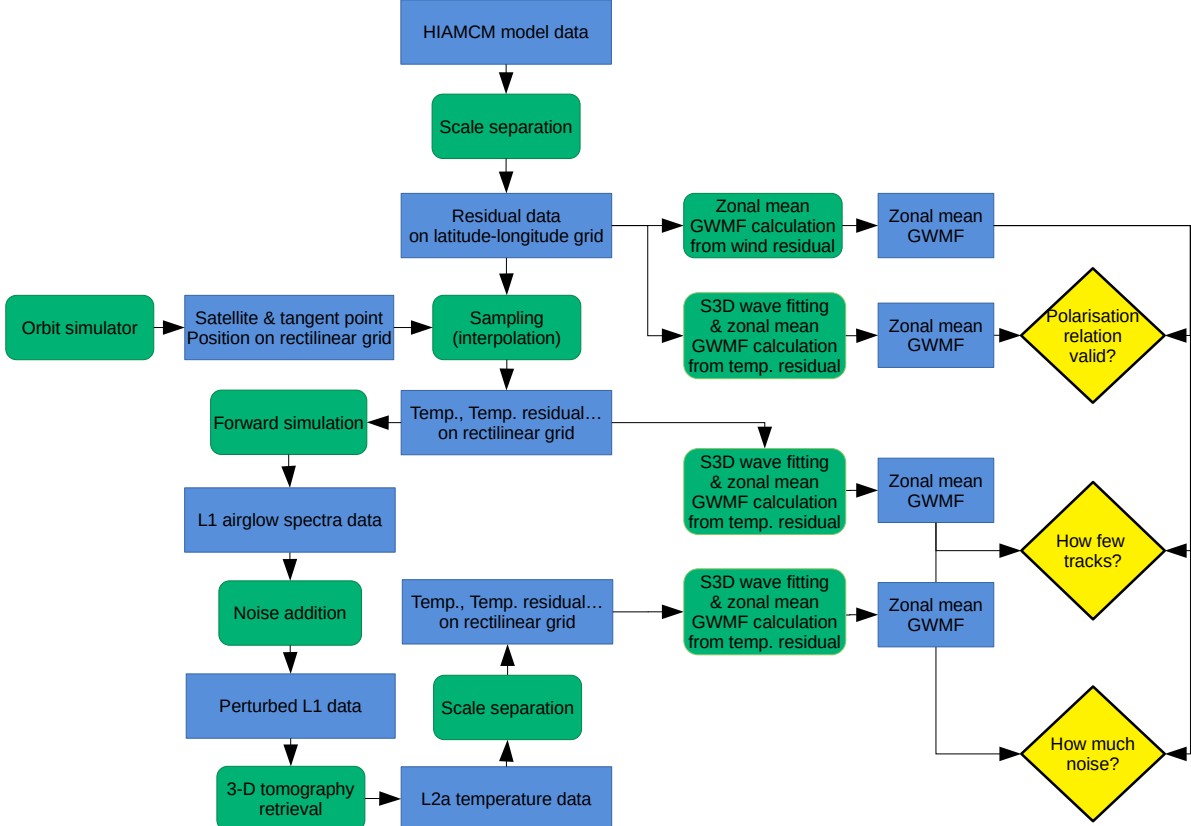

**Figure 2.** Schematic flow diagram of the assessment steps to address the three major questions highlighted in yellow diamond boxes. The assessment is based on comparison with zonal mean GWMF calculated directly from the model winds and considered here as the reference of truth. From top to down further assumptions and/or constraints are added as the tested data become more similar to the real observations.

## 2.3 Method of assessment

The E2E approach for evaluating the performance of the proposed mission concept is illustrated in Fig. 2. We employ HIAMCM model data (Becker and Vadas, 2020, see Sect. 3.1) for a realistic, self-consistent basis of the E2E simulations. The distributions of temperature, density, and wind velocities are separated for large scale structures such as planetary waves and tides on the one hand and small scale residuals due to GWs on the other hand. We consider both wind and temperature data. Temperatures are the observation target of the measurement method. From the winds we gain our reference for the assessment. As described

in the introduction, zonal mean GWMF calculated directly from the wind perturbations provides an unambiguous reference of truth against which we can compare the values from simulated observations and thus quantify the influence of the various assumptions or constraints needed for observing GWMF with a real instrument.





The first question which we address is the applicability of the polarization relation for the MLT. GWMF can be deduced from 3D temperature data by determining the 3D wave vector assuming polarization and dispersion relations (cf. Sect. 4.1).

In order to test the validity of this approach, we apply our wave analysis (S3D; Sect. 3.7) directly to the full model data and compare thus generated zonal mean GWMF with our reference.

The second question regards the number of tracks required for reliable GWMF quantification. For this assessment, we sample model residual temperatures onto various synthetic measurement geometries for different numbers of observation tracks, apply S3D and, again, compare against the reference.

Third, we assess the influence of the observation technique which comprises both the observational filter as a general limitation of limb sounding as well as the specific instrument parameters. This assessment therefore encompasses all steps of observation starting from synthetic radiances generated by forward modeling, a simplified instrument model adding realistic noise and a tomographic retrieval. Again S3D is applied to the outcome of the full E2E simulations and zonal mean GWMF is assessed against the reference. Different noise levels are tested in order to determine the robustness against instrument

performance.

## 3 Models, tools and instrument

This section describes the atmospheric circulation model and the radiative transfer model we base our study on, introduces the SHS instrument proposed for the observation and gives an introduction to the wave analysis and retrieval tools employed.

### 3.1 HIAMCM

The HIAMCM (HIgh Altitude Mechanistic general Circulation Model) (Becker and Vadas, 2020; Becker et al., 2022) is a new, high-altitude version of the KMCM (Kühlungsborn Mechanistic general Circulation Model). The HIAMCM employs a spectral dynamical core with a terrain-following hybrid vertical coordinate. This includes a correction for non-hydrostatic dynamics and physically consistent thermodynamics in the thermosphere. The current model version uses a triangular spectral truncation at a total wavenumber of 256, which corresponds to a horizontal grid spacing of $\sim 52$ km. The altitude-dependent

vertical resolution includes 280 full levels. The level spacing is dz $\sim 600$ m below z $\sim 130$ km, with dz increasing with altitude farther above and dz $\sim 5$ km above z $\sim 300$ km.

The HIAMCM captures atmospheric dynamics from the surface to approximately 450 km. GWs are simulated explicitly with an effective resolution that corresponds to a horizontal wavelength of $\sim 200$ km. Non-resolved scales are parameterized by macro-turbulent vertical and horizontal diffusion based on the Smagorinsky model. Since molecular viscosity is taken

into account for both vertical and horizontal diffusion, the HIAMCM does not require an artificial sponge layer. Resolved GWs are dissipated self-consistently by molecular diffusion in the thermosphere above $\sim 200$ km, and predominantly by macro-turbulent diffusion at lower altitudes. These features allow the HIAMCM to capture the generation, propagation, and dissipation of medium-scale GWs, including their interactions with the large-scale flow and the generation of secondary and





tertiary waves. This capability is essential to simulate GW dynamics in the MLT (Becker and Vadas, 2018) and at higher
altitudes (Vadas and Becker, 2019).

The HIAMCM employs radiation and moist convection schemes that are simplified compared to comprehensive methods.
Furthermore, the model does not include a chemistry module, and ion drag is the only ionospheric process that is accounted
for. To distinguish these idealizations from methods employed in community models, the HIAMCM is said to be a mechanistic
model. Nevertheless, the key features of a climate model (topography, simple ocean model, radiative transfer, boundary layer
processes, tropospheric moisture cycle) are fully taken into account. Also note that the HIAMCM is currently the only GW-
resolving whole atmosphere model that can be nudged to reanalysis in the troposphere and stratosphere, allowing for the
simulation of observed events (Becker et al., 2022).

## 3.2   Orbit simulator

The simulation of the orbit (illustrated in Fig. 3) is based on a fixed orbit inclination, orbit altitude (shown in Table 1) and
start longitude at the beginning of the day. Assuming a spherical Earth and constant gravity acceleration scaled to orbit altitude
the in-orbit velocity is determined. A time series of orbit positions on Earth surface is calculated from the satellite position on
the orbit fixed in space and the rotation of the Earth. A grid for atmosphere representation (atm-grid in the following) is then
generated based on the tangent points of backward viewing direction for $80\,\mathrm{km}$ altitude and spanning a local rectilinear grid
with $x$-direction along this tangent point track, a $y$-direction perpendicular to this tangent point track and the local vertical.
Both satellite position and atm-grid are then used to build a complete set of matching tangent points for radiative transfer and
retrieval simulations as described in in Sect. 3.3 &  3.6. The basis for the radiative transfer simulations are the atmospheric
quantities such as temperature and pressure interpolated from the longitude-latitude grid of the general circulation model to the
atm-grid by means of spline interpolation.

**Table 1.** Orbit parameters used for the simulated observations

| Parameter | Property |
| --- | --- |
| Orbit altitude | $500\,\mathrm{km}$ |
| Orbit inclination | $97.3°$ |
| View direction of center track | backward with respect to flight vector |

## 3.3   O$_2$ A-band airglow photochemistry and radiative transfer process

This section describes the photo-chemical processes which initiate the generation of the O$_2$ A-band emission, followed by a
short description of radiative transfer process propagating the radiance to the instrument.

O$_2$ can be present as $^{16}$O$_2$, $^{17}$O$^{16}$O or $^{18}$O$^{16}$O, where the latter two can be neglected due to their low abundance, as shown
by Slanger et al. (1997). In general, an excited molecule can be in one of multiple electronic states, where for each electronic
state the radicals can be in one of multiple vibrational states. The transitions between different electronic-vibrational states


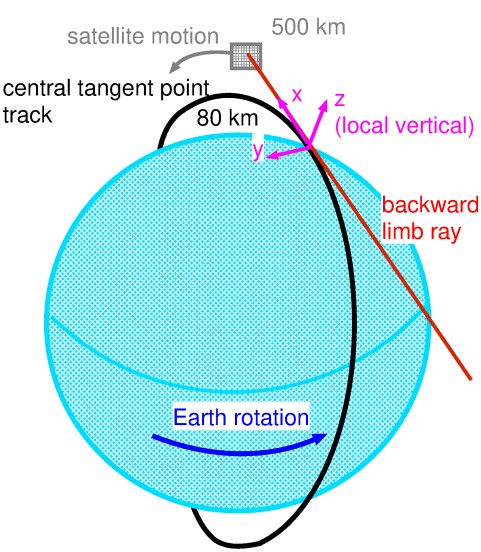

**Figure 3.** Schematic view of the synthetic orbit and grid generation.

form atmospheric emission and absorption bands. Each band consists of multiple emission lines due to the transitions within

multiple rotational states of vibrational states. We measure the A-band emission at $762\,\mathrm{nm}$, which is the electronic transition

from the second excited state $O_2(b^1\Sigma_g^+, v{=}0)$ to the ground state $O_2(X^3\Sigma_g^-, v{=}0)$. A detailed description of the dayglow $O_2$

A-band is given by Sheese (2009), Bucholtz et al. (1986) and Zarboo et al. (2018). It has three sources which are depicted in

Fig. 4a.





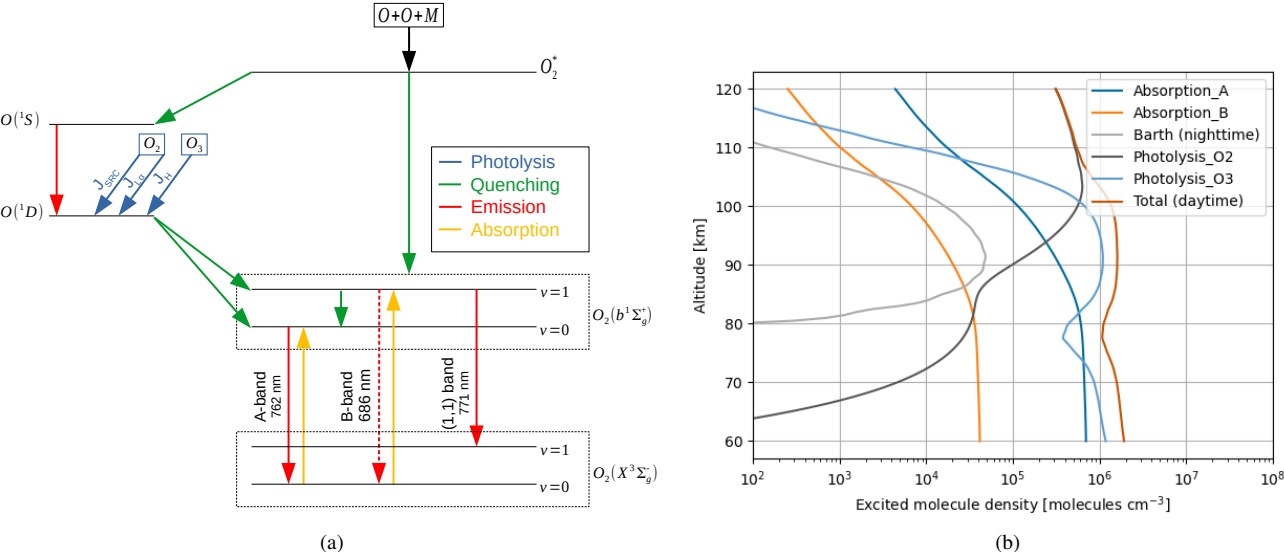

(a) (b)

**Figure 4.** (a) Schematic of production of excited state $O_2(b^1\Sigma_g^+, v=0)$; dashed lines indicate neglected transitions; (b) number density of excited $O_2$ molecules due to the different production mechanisms using HAMMONIA model; note that during daytime all five production mechanisms are active, whereas during nighttime only the Barth process is active;

First, the excited state can be produced by photon absorption in the atmospheric bands. Bucholtz et al. (1986) show that the $\gamma$-band absorption from $O_2(X^3\Sigma_g^-, v=0)$ to $O_2(b^1\Sigma_g^+, v=2)$ can be neglected and hence is not shown in Fig. 4a. Only the absorption in the A- and B-bands is therefore considered. The excited molecules in $O_2(b^1\Sigma_g^+, v=1)$ are rapidly deactivated to $O_2(b^1\Sigma_g^+, v=0)$ via a quenching process. Slanger et al. (1997) argue that the B-band emission at $686\,\text{nm}$ is insignificant compared to the quenching process, and thus can be neglected. The emission in the $(1, 1)$ band is considered. The second source

is due to photolysis of $O_2$ in the Schumann-Runge continuum ($J_{SRC}$) and at Lyman $\alpha$ ($J_{L\alpha}$) and due to the photolysis of $O_3$ in the Hartley band ($J_H$). It produces excited atomic oxygen $O(^1D)$, which transforms to $O_2(b^1\Sigma_g^+)$ due to collisional excitation with $O_2$ in ground state. The third source is a chemical source, which is independent of solar radiation and thus also present during night time. The source of this process is a three body recombination of atomic oxygen, producing electronically exited $O_2^*$ radicals. From there, $O_2(b^1\Sigma_g^+)$ is produced through a direct quenching process or a chain of quenching processes, going

through $O(^1S)$ and $O(^1D)$. This process was first described by Barth and Hildebrandt (1961) and thus, is called Barth process. Since some of the related rate coefficients are not well known, McDade et al. (1986) proposed a model with fitting parameter to describe the Barth process. A detailed description of the calculation of $O_2$ A-band emission is given in Appx. B. Fig. 4b shows the number density of excited $O_2$ molecules due to the different production mechanisms using HAMMONIA model data. Note that during daytime all five production mechanisms are active, whereas during nighttime only the Barth process is active.

The $O_2$ A-band airglow emissions are transmitted through the atmosphere before they are detected by an instrument. The observed airglow spectra are integrated slant-path radiances along the instrument viewing line of sight (LOS). For limb ob-



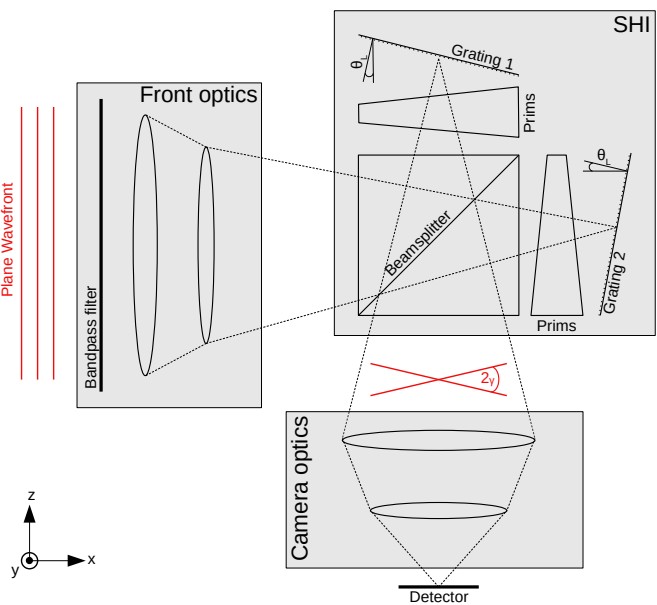

**Figure 5.** Schematic of the SHS instrument

servations, the emissions from the lowermost tangent points along the LOS contribute the most to the integrated radiances provided that the atmosphere is still optically thin for the emission lines.

Due to the high abundance of ground-state $O_2$ molecule in the atmosphere and its self-absorption effect, the emitted $O_2$
A-band radiance can only partly pass through the atmosphere and reach the instrument detector. This radiative transfer process is described by the Lambert-Beer law, and the observed spectral irradiance intensity can be written in integral form as:

$$I(\nu) = \int\limits_{-\infty}^{\infty} I(s)D(\nu,s)\exp\Big(-\int\limits_{s}^{\infty} n(s')\sigma(s')D(\nu,s')ds'\Big)ds, \tag{2}$$

where $\nu$ refers to the wavenumber of the spectral line, $s$ denotes the LOS distance, $D(\nu)$ represents the line shape broadening profile, which is dominated by Doppler broadening in the middle and upper atmosphere, $n(s)$ is the number density of $O_2$, $\sigma$
is the $O_2$ absorption cross section.

In the lower atmosphere, nearly all $O_2$ A-band emissions are self-absorbed. Thus, $O_2$ A-band airglow cannot be detected on ground. In the altitude region of our interest, from $60\,\mathrm{km}$ to $120\,\mathrm{km}$, we have assumed an optically thick conditions for the center wavelength of the emission lines .

### 3.4 Spatial heterodyne interferometer

A spatial heterodyne interferometer is based on the principle of a Michelson interferometer, but where the two mirrors are replaced by fixed tilted gratings. The simplified concept of the SHS is shown in Fig. 5. Light of frequency within the spectral bandpass enters the instrument along the optical axis. Light of each frequency is split into two waves by the beam splitter





and diffracted and reflected by the gratings. The two waves arrive back to the beam splitter producing an interference pattern, which is forwarded to the detector. Considering multiple emissions within the bandpass, an interferogram consists of multiple

superposed interference patterns which can be described by sinusoidal waves. The spatial frequency of the produced fringe pattern follows from the grating equation, denoted by

$$\sigma(\sin\theta_L + \sin(\theta_L - \gamma)) = \frac{m}{d}, \tag{3}$$

where $\sigma$ is the wavenumber of the incoming light, $\theta_L$ the blaze angle of the gratings, also called Littrow angle, $d^{-1}$ is the grating groove density, $m$ is the diffraction order and $\gamma$ the outgoing diffraction angle with respect to the Littrow angle. The

Littrow wavenumber is the wavenumber where $\gamma = 0$, thus

$$\sigma_L = \frac{m}{2\sin(\theta_L)d}. \tag{4}$$

Using Taylor expansion, Harlander et al. (1992) show that the spatial frequency is dependent on the wavenumber by

$$f(\sigma) = 4(\sigma - \sigma_L)\tan(\theta_L)M, \tag{5}$$

where $M$ is the magnification factor, introduced by the camera optics.

This equation shows that the the relation between spatial frequency and wavenumber is symmetric around the Littrow wavenumber.

Following Roesler and Harlander (1990) and Deiml (2018), an ideal one-dimensional interferogram along the x-axis can be described by

$$I(x) = \int_{b_0}^{b_1} B(\sigma)\left[1 + \cos(2\pi f(\sigma)x)\right]d\sigma, \tag{6}$$

where $B$ is the spectral radiance and $b_0$ and $b_1$ are the lower and upper bound of the spectral filter, respectively.

The interferogram is transformed into a spectrum by Fourier transformation. Fig. 6 shows the $O_2$ A-band emission for two temperatures and the corresponding spectra as seen by the instrument. The temperature dependency as seen in the lower panel of Fig. 6 is used to retrieve temperature.

A silicon-based detector is used in this instrument. The operation in ambient to cool condition give a shot noise limited

system for integration times of 1–10 sec (Liu et al. (2019)). Shot noise can be modelled by a Poisson process with mean and variance equal to the signal. For a signal above 10 counts, the Poisson distribution approximates a normal distribution about its mean. Thus, for simplicity the shot noise is approximated by an additive white Gaussian noise with standard deviation equal to the square root of the signal in each pixel.

### 3.5 Split of the interferogram

Revisiting Fig. 5, the front optics map the rays of one point in the object plane onto one point in the image plane on the gratings. The gratings induce a path length difference which leads to constructive and destructive interference at the associated position.



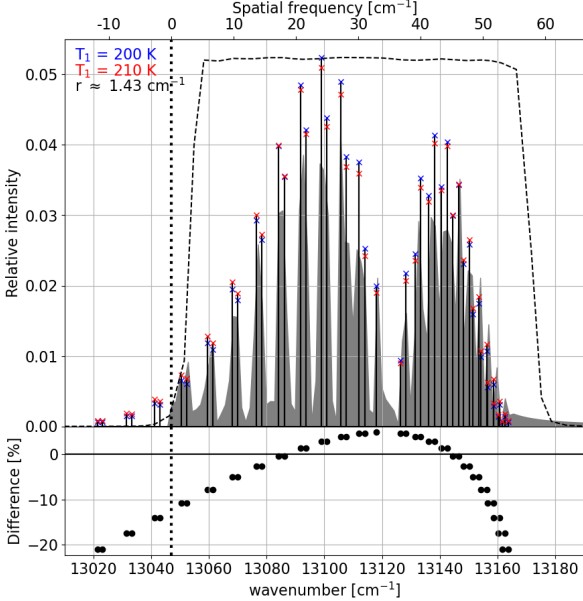

**Figure 6.** Modelled $O_2$A-band seen by the SHI instrument for temperature at $200\,K$ and $210\,K$; the spectra are presented by the Dirac impulses such that the sum over all emission line is equal to 1; below the relative intensity difference between the spectrum at $200\,K$ and $210\,K$ is shown; The dashed line shows the theoretical filter curve and the dotted vertical line the designed Littrow wavenumber of the AtmoLITE instrument; the grey shaded area shows the spectrum convolved by the instrument line shape; $r= 1.43cm^{-1}$ is the spectral step size.

Accordingly, the SHI performs a spatial mapping of the atmospheric scene onto the detector. Each side of the interferogram contains spectral information of the associated region within the atmospheric scene which enables a split of the interferogram at zero optical path difference (ZOPD). This can be employed to derive two independent temperatures along the horizontal

(spatially and spectrally superimposed) axis. The concept to mirror the interferogram at the ZOPD has been already used by Johnson et al. (1996) for the far-infrared spectrometer (FIRS)-2 and by Gisi et al. (2012) for the TCCON FTIR spectrometer to gain a higher resolution. However, Ben-David and Ifarraguerri (2002) and Brault (1987) point out that the phase correction including finding the correct ZOPD is crucial.

The following simulation demonstrates that an averaged temperature can be recovered from an analytical inteferogram.

We assume a linear temperature gradient across the horizontal field. We simulate each pixel with the associated temperature and assemble the full interferogram pixel by pixel. Subsequently, the interferogram is split at the ZOPD and each side is symmetrically extended to get two full interferograms. Each side can be then used to retrieve a temperature. Fig. 7 shows the simulation result of an ideal interferogram without noise using a simple temperature retrieval which minimises the squares of the residuals. The red circle and the blue diamond in Fig. 7a indicate the retrieved temperature, which are about $17\,km$ apart for





the given specifications. Note that the two retrieved temperatures are dragged towards the center, because most of the intensity of the interferogram is around the ZOPD, which is spread over the entire spectrum after Fourier transformation is applied. This allows theoretically to acquire two independent cross track measurements with one instrument. An in depth validation using calibration and orbit data considering noise and instrument errors is content of future research.

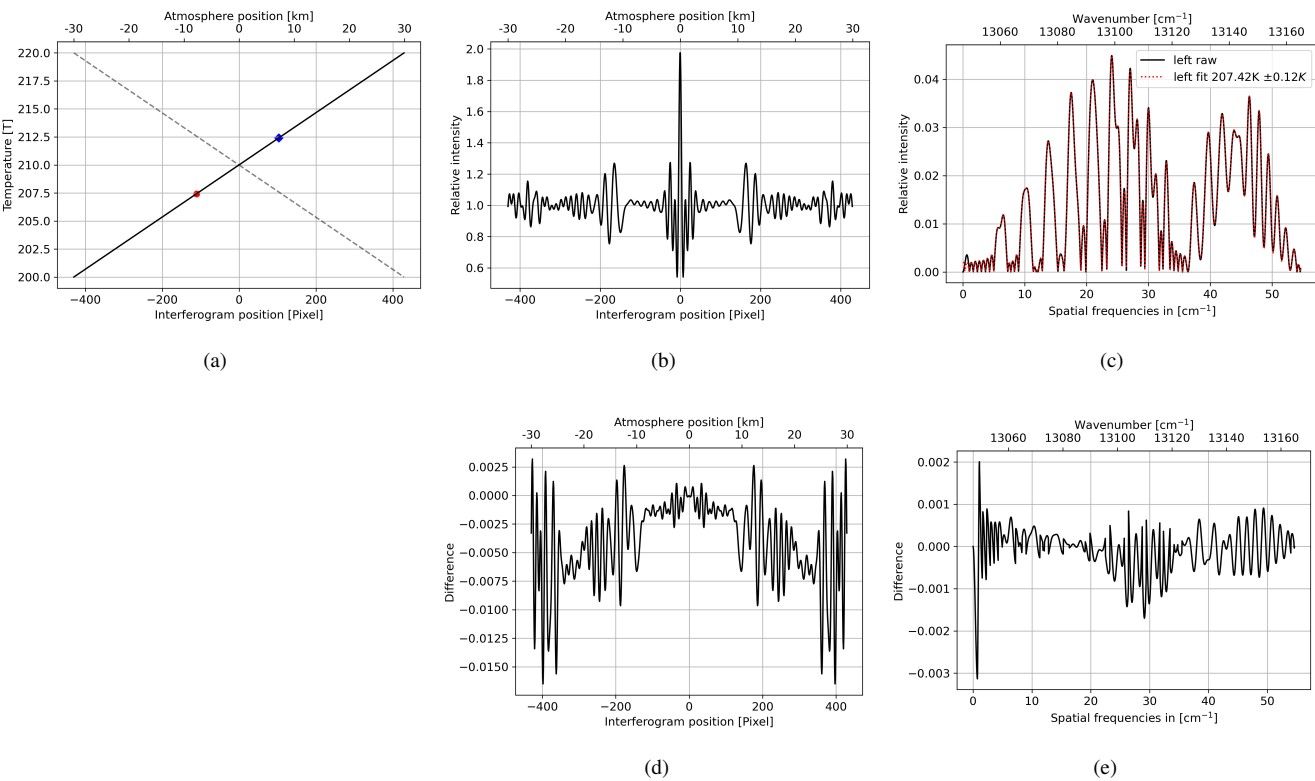

**Figure 7.** Simulation of interferogram split assuming a linear temperature gradient; (a) shows the temperature gradient from 200 K to 220 K across the horizontal field (solid line) where each side is symmetrically extended around the center (dashed lines); the red circle and the blue diamond indicate the location of the retrieved temperature for the the left and right hand side; (b) shows the left symmetrically extended interferogram; (c) shows the temperature retrieval of (b); (d) shows the difference between the left and right symmetrically extended interferogram; (g) shows the difference between the left and right symmetrically extended raw spectrum;

### 3.6 Tomographic retrieval for generating 3D atmospheric volumes

Retrieving temperatures from measured limb spectra is a classic inverse problem. That is, we have a radiative transfer model that can compute measured spectra from an assumed atmospheric state (forward model), but the inverse problem is much harder, since it is typically both underdetermined (multiple atmospheric states could result in the same set of measurements) and ill-posed (there might be, in theory, no atmospheric states that could result in a given real-life measurement affected by instrument noise and other error sources). Such a problem is solved using the forward model and a mathematical framework





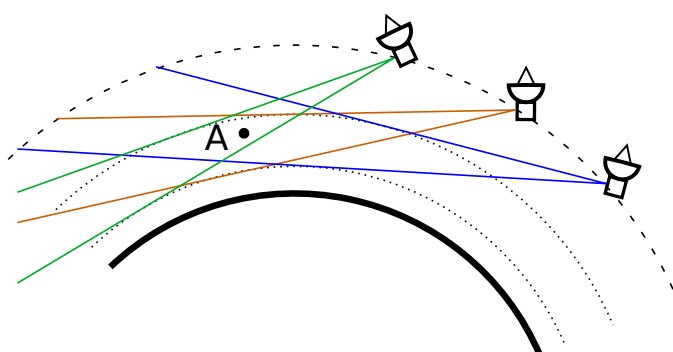

**Figure 8.** The principle of 2D tomography. The field of view of the satellite within the orbital plane is shown for different satellite positions along the orbit. A point A inside the airglow layer lies within the field of view for all three positions, therefore airglow radiance at point A contributes to observed radiances at all these positions and every position in between. We can use the observations to solve for airglow radiance at point A (and all other such points) using inverse modelling.

for inverse modelling (e.g. Rodgers, 2000). The main idea of this approach is iterative minimisation of the following function (called cost function):

$$J\left(\boldsymbol{x}\right) = \left(\boldsymbol{y} - \boldsymbol{F}\left(\boldsymbol{x}\right)\right)^{T}\mathbf{S}_{\epsilon}^{-1}\left(\boldsymbol{y} - \boldsymbol{F}\left(\boldsymbol{x}\right)\right) + \left(\boldsymbol{x} - \boldsymbol{x}_{\mathrm{apr}}\right)^{T}\mathbf{S}_{\mathrm{a}}^{-1}\left(\boldsymbol{x} - \boldsymbol{x}_{\mathrm{apr}}\right). \tag{7}$$

Here $\boldsymbol{y}$ is a set of measurements taken by the instrument, $\boldsymbol{x}$ is the candidate atmospheric state. $\boldsymbol{F}(\boldsymbol{x})$ is the forward model, it maps an atmospheric state $\boldsymbol{x}$ to the set of measurements that the instrument would acquire if the atmosphere was indeed

in the state $\boldsymbol{x}$. $\boldsymbol{x}_{\mathrm{apr}}$ represents our prior knowledge of the atmospheric state. In our case that is an estimate of the large-scale temperature structure of the atmosphere without gravity waves. $\mathbf{S}_{\epsilon}^{-1}$ and $\mathbf{S}_{\mathrm{a}}^{-1}$ are positive-definite symmetric matrices (covariance matrices).

The first term on the right hand side of equation (7) quantifies how closely the atmospheric state $\boldsymbol{x}$ matches the observations $\boldsymbol{y}$. One can construct covariance matrix $\mathbf{S}_{\epsilon}^{-1}$ knowing the measurement error characteristics typical for the instrument in

question. The second term quantifies how likely the atmospheric state $\boldsymbol{x}$ is given our prior knowledge about the atmosphere. This knowledge includes both the base state of the atmosphere $\boldsymbol{x}_{\mathrm{apr}}$, and more general considerations (such as, for example, that large spatial discontinuities in temperature are unlikely) that are taken into account when constructing $\mathbf{S}_{\mathrm{a}}^{-1}$ (Ungermann et al., 2010a). Equation (7) is solved iteratively using the Levenberg–Marquardt algorithm (Marquardt, 1963) and a conjugate gradients solver. For both the forward model and the inverse modelling we need a computationally efficient implementation.

Here, we employ the JURASSIC2 forward model (e.g. Ungermann, 2013) to simulate the spectra based on a 2-D discretisation of the atmosphere along the satellite track. This model was enhanced for this study by a simple adjoint line-by-line model dedicated to the simulation of the $O_2$ A-band (see Sec. 3.3). The inversion uses the JUTIL python library to bring the synthetic and measured spectra in agreement using a truncated conjugate gradient trust region method (e.g. Ungermann et al., 2015).





In order to perform a full end-to-end test, noise is added to simulated observations to emulate instrument performance.
To generate the noise in the synthetic measured spectra, we compute from the spectrum the number of photons hitting (on average) one detector pixel per second and use this number to determine Gaussian noise assuming that the instrument is shot-noise limited and the dark current is negligible; i.e., the 1-$\sigma$ is computed from the square root of number of photons. Note that we assume an efficiency of 0.2, i.e. only one fifth of inbound photons will end up in the modulated part of the interferogram. This noise is then reduced according to assumed averaging in time and space ($4.5\,\text{s}$ integration time, 13 vertical detector rows, 4 horizontal detector columns).

The individual measurement tracks of the proposed satellites are treated separately from one another and each constitute a separate 2-D slice at the orbital plane of the satellite or very close to it (Figure 8). Limb measurement geometry and backward viewing direction result in each air parcel within the 2-D slice being observed from multiple points of the satellite orbit. This allows to reconstruct 2-D temperature cross-section in a tomographic fashion. The full 3-D state is reassembled afterwards from the individually retrieved 2-D cross-sections .

As explained in Sect. 3.3 there are more excitation mechanisms and higher emissions during daytime than during nighttime. For one retrieval, we therefore split the orbit at the position of the terminator in roughly two equal halves, one containing the daytime, the other containing the night time measurements (even though we currently do apply the same retrieval settings to both). We then lengthen the two orbit parts by an additional $\approx 1500\,\text{km}$ on both ends. Horizontally, a sampling of $30\,\text{km}$ is used, while the vertical range is sampled in $1\,\text{km}$ steps from $60\,\text{km}$ to $120\,\text{km}$ and in $2\,\text{km}$ steps above (Table 2).

The forward simulations employ a regular grid of two rays per detector row, which are combined assuming a line spread function of Gaussian shape with an 1-$\sigma$ corresponding to the height of the row.

The spectra range from $13\,060\,\text{cm}^{-1}$ to $13\,160\,\text{cm}^{-1}$ with a sampling distance of $2\,\text{cm}^{-1}$ and a spectral resolution (FWHM) of $\approx 3.9\,\text{cm}^{-1}$ due to the employed strong Norton-Beer apodisation (Norton and Beer, 1976). More details about the simplifying assumptions for the forward model in the tomographic retrieval are given in Appx. C.

### 3.7  JUWAVE S3D wave analysis

In this study we investigate GWs in a narrow stripe (small swath width) of observations along the tangent point tracks. This requires an analysis method which can, at least in one direction, analyze waves with notably longer wavelengths than the size of the analysis volume. In addition, the vertical wavelength of gravity waves is refracted by the background wind, which contradicts the assumption of a stationary wave spectrum over the range of the analysis volume, which is made e.g. by Fourier transform. Therefore, the small volume sinusoidal fit method (S3D) was developed and tested for the purpose of GW analysis in small observation volumes and for highly localized GW fields (Lehmann et al., 2012).

In this method, the observation volume or model domain is divided into small sub-volumes and a sinusoidal fit performed on each sub-volume:

$$T_i' = \sum_j A_j \sin(\boldsymbol{k_j}\boldsymbol{x_i}) + B_j \cos(\boldsymbol{k_j}\boldsymbol{x_i}), \tag{8}$$





**Table 2.** Summary of detector and sampling properties

| Parameter | Property |
| --- | --- |
| Detector columns/rows | 800/800 |
| Etendue | $0.01\,\mathrm{cm^2 sr}$ |
| Efficiency | 0.2 |
| Integration time | $4.5\,\mathrm{s}$ |
| Averaged rows | 13 |
| Averaged columns | 4 |
| Spectral range | $13\,060\,\mathrm{cm^{-1}}$ to $13\,160\,\mathrm{cm^{-1}}$ |
| Spectral sampling | $2\,\mathrm{cm^{-1}}$ |
| Spectral resolution | $3.9\,\mathrm{cm^{-1}}$ |
| Lowest tangent altitude | $60\,\mathrm{km}$ |
| Highest tangent altitude | $120\,\mathrm{km}$ |
| Tangent altitude spacing | $1\,\mathrm{km}$ |
| Vertical sampling (below $120\,\mathrm{km}$) | $1\,\mathrm{km}$ |
| Vertical sampling (above $120\,\mathrm{km}$) | $2\,\mathrm{km}$ |
| Horizontal sampling | $30\,\mathrm{km}$ |

where $T_i'$ is the temperature fluctuation at the location $\boldsymbol{x_i}$ and $\boldsymbol{k_j x_i}$ is the scalar product between the wave vector of the $j$'s wave component with the spatial coordinate vector of the $i$'s point in the analysis volume. The wave components $j$ are determined sequentially, subtracting the wave field of component $j$ from the fit volume before fitting $j+1$. In this study, three wave components are fitted. Amplitudes and wave vectors are determined via least squares fit: amplitudes are determined analytically, wave vectors via a variational approach; the minimum $\chi^2$ from a steepest descent method and a nested interval method is selected.

The size of the analysis cube is selected in a way that most of the spectral content has wavelength of $cs_d/2 < \lambda_d < 3cs_d$, with $\lambda_d$ being either the horizontal or the vertical wavelength and $cs_d$ the analysis volume diameter. This choice is motivated by previous sensitivity studies (Preusse et al., 2012) and will be discussed further in Sect. 4.2.1. In particular, we find that a vertical cube size of $15\,\mathrm{km}$ comprises the spectral power in the MLT region almost entirely. In the lower thermosphere, which is analyzed for consistency reasons as well, larger cube sizes are needed. To retain only reliable fits, we omit all fits with derived wavelengths larger than 3 times the cube size from evaluation. In order to enhance the vertical resolution and hence to better capture the loss of GWMF by the approach to a critical level at the MLT wind reversal, a refit of amplitude and phase only based on the wavevector from the initial fit is performed. For this study, we keep the horizontal cube size the same but reduce the vertical cube size to $5\,\mathrm{km}$, with the exception of Sec. 4.1, where the results are obtained through a vertical cube size reduction to $4\,\mathrm{km}$. In the same section, the initial cube grid consists of cubes with sizes $300\,\mathrm{km}\times300\,\mathrm{km}\times15\,\mathrm{km}$ (at $75\,\mathrm{km}$ altitude) and $600\,\mathrm{km}\times600\,\mathrm{km}\times20\,\mathrm{km}$ (at $130\,\mathrm{km}$ altitude).





The cube size is defined via the number of analysis points for the synthetic observation data, which have a fixed sampling in $x$, $y$ and $z$ direction. For the model data, a fixed model sampling in longitude direction means, distance-wise, a coarser

sampling close to the equator and a finer sampling at high latitudes. Therefore, a fixed cube size in kilometers is specified and the number of fitting points adapted accordingly. This ensures that the same part of the spectrum is targeted independent of latitude.

The results of S3D are expected to be a good compromise between spatial and spectral representation. It has been shown by (Lehmann et al., 2012) that the spectral distribution composed of all S3D wave fits in a given region well reproduces the spectral

content obtained via Fourier analysis of the same region. At the same time, waves are well localized and can be attributed to individual source features.

## 4    Assessment

In this section we use the methods described in Sect. 3 and follow the assessment approach outlined in Sect. 2 in order to quantify to which accuracy GWMF can be inferred from MLT limb observations and how many independent across-track

points (i.e. how many measurement tracks) are required. The assessment measure is the comparison of global GW momentum flux values from simulated temperature observations to the values directly obtained from wind fluctuations of the full model fields. In addition, we consider how well spectral information is conserved when reducing the number of observation tracks. The assessment starts from consistent fluctuations in temperature and wind velocities after scale separation applied to a single snapshot of full model data (cf. Sect. 2.1), i.e. the scale separation is applied but not subject to the assessment.

### 4.1    Polarization relations

In the proposed mission concept GWMF is inferred from 3D temperature structures. This requires that a) polarization relations are valid in the MLT also under non-linear conditions of many GWs approaching a critical level and that b) a few-wave decomposition approach as S3D is an adequate method for determining the 3D wave vectors of the leading GWs. This is tested here using the approach from Sect. 2.3.





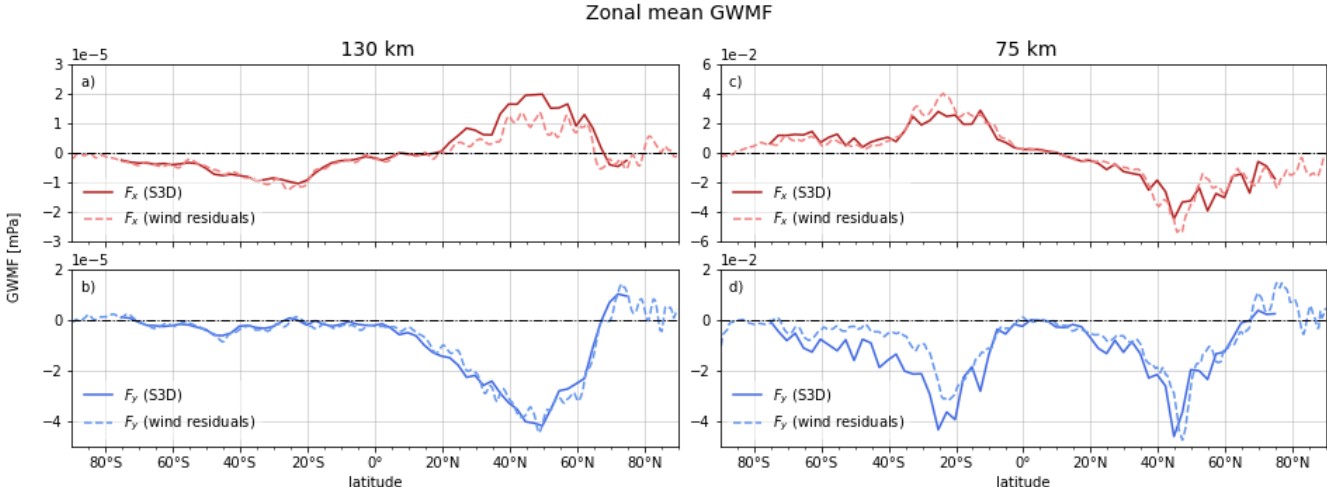

**Figure 9.** The zonal mean vertical flux of zonal (red) and meridional (blue) momentum. Dashed lines show GWMF calculated from wind residuals [Eq. 9], while solid lines are calculated using wave vectors and residual temperature amplitudes [Eq. 10]. Left column shows the fluxes at $130\,\mathrm{km}$ altitude while the right column shows fluxes at $75\,\mathrm{km}$. Wind based fluxes have been smoothed by averaging over $4\,\mathrm{km}$ altitude bins, centered at the specified altitude, and by averaging over $5°$ latitude bins.

In order to verify that the momentum flux based on the S3D-derived wave parameters can be correctly calculated, a comparison with momentum flux estimates directly from wind residuals is carried out. As outlined in Sect. 2.2, we use zonal mean GWMF (without the correction for Coriolis force) as true reference:

$$(F_x, F_y) = \rho\left(\overline{u'w'}, \overline{v'w'}\right), \tag{9}$$

where $F_x$ and $F_y$ are the zonal and meridional components, respectively. Following the derivation of the vertical flux of
horizontal gravity wave *pseudo*momentum by (Ern et al., 2004), we express Eq. 9 in terms of the residual temperature wave amplitude $\hat{T}$, wavenumbers $k$, $l$, $m$, and intrinsic frequency $\hat{\omega}$, as

$$(F_x, F_y) = \frac{1}{2}\rho\left(\frac{g}{N}\right)^2 \left(\frac{\hat{T}}{T}\right)^2 \frac{(k,l)}{m} \times \left(1 - \frac{f^2}{\hat{\omega}^2}\right)^{-1}. \tag{10}$$

In Eq. 10, $N$ is the buoyancy frequency, $T$ is the background temperature and the factor $(1 - f^2/\hat{\omega}^2)^{-1}$ converts from pseudomomentum to momentum. Equation 10 is a simplified expression omitting two correction terms which are discussed in
the supporting material to Ern et al. (2017) and relevant only to high frequency GWs not considered in this study. The wave parameters of Eq. 10 are acquired through S3D analysis on temperature residuals of the HIAMCM data.

     Figures 9a and 9b show the zonal and meridional components of GWMF at an altitude of $130\,\mathrm{km}$. The temperature-derived flux, based on wave parameters from global evenly distributed S3D analysis volumes, is in good agreement with the wind-based method used for reference. There is dissimilarity in the zonal GWMF at around $50°\mathrm{N}$, indicating a region of strong wind
shear and/or a breakdown of the validity of the linear approximation. Likewise, at lower altitude the methods are consistent,



with some differences in the zonal component around $25°$S (Fig. 9c) and in the meridional component across the southern hemisphere midlatitudes (Fig. 9.d), suggesting dynamics that are not completely captured. As a whole, the results from the two different methods are in good agreement and confirm that Eq. 10, based on wave properties from S3D analysis, is suitable for studying wave dynamics in the MLT.

## 4.2 How few tracks are required?

One of the major questions concerning our approach is how many measurement tracks are necessary to sufficiently derive the GW parameters in the MLT region within the frame of proposed two-CubeSat observation strategy. For this purpose we test the E2E simulations with a series of varying number of tracks and swath widths. The detailed results are presented in this section with a focus on the interpretation and intercomparison of GW wave vectors and zonal mean momentum flux values. All results presented in this subsection are either from sampled orbit data or full E2E simulations.

### 4.2.1 Analysis of wavelength spectra

Spectra of PGWMF as function of horizontal and vertical wavelengths are considered for two reasons. First, the choice of the cube size restricts the long-wavelength limit of the analyzed spectrum. As described in Sect. 3.7, the desired wavelengths should ideally be in a range of [1/2, 3] times the cube size, wavelengths larger than 3 times the cube size are rejected. We hence need to verify that our cube-size choice does not cut-off major parts of the spectrum. Second, the wavelength spectrum should remain (largely) unchanged when reducing the number of tracks.

We start with what we deem a good initial value for the cube size: From the model set-up we expect shortest horizontal wavelengths of O(200 km). Based on previous experience and also using a separation scale of zonal wavenumber 18, longest wavelengths to be considered are O(2000 km). An average vertical wavelength around 12 km was found from SABER data (Ern et al., 2018) for 80 km altitude. Therefore an initial cube size of 600 km along-track $\times$ 420 km across track $\times$ 15 km altitude is selected for fitting the wave vectors and a reduced vertical size of 5 km for refitting the wave amplitudes. With an atm-grid sampling of 30 km $\times$ 30 km $\times$ 1 km (along-track $\times$ across-track $\times$ vertical; cf. Sect. 3.6), this corresponds to to 21 $\times$ 15 $\times$ 15 points. Spectra in terms of horizontal and vertical wavelengths for these initial cube size are shown in Fig. 10a-c for altitudes of 75, 85 and 95 km.

The spectral peak appears at around 600 to 800 km horizontally and 10 to 16 km vertically at all altitudes. All spectra are cut off at longer wavelength of around 2500 km horizontally and 45 km vertically, as the detection upper limit. It results from the limits when filtering reliable fits, which are up to 3.5 (horizontally) and 3 (vertically) times the cube sizes. This, however, does not remove major parts of the spectrum. The spectrum is slightly truncated at shorter horizontal wavelength around 140 km at an altitude of 75 km, which is not the case for 95 km as waves with longer wavelengths can propagate higher. These wavelength spectra confirm our expectation that the target spectral range is well covered by the selected cube size parameters.

In order to investigate the impact of fewer tracks, the number of measurement tracks is reduced in several steps. The proposed mission deploys 2 SHI, both with split interferograms to provides in total 4 measurement tracks. The distance of the track pair from one SHI is assumed to be the distance of the geometric center of the half-interferogram (30 km). The distance between



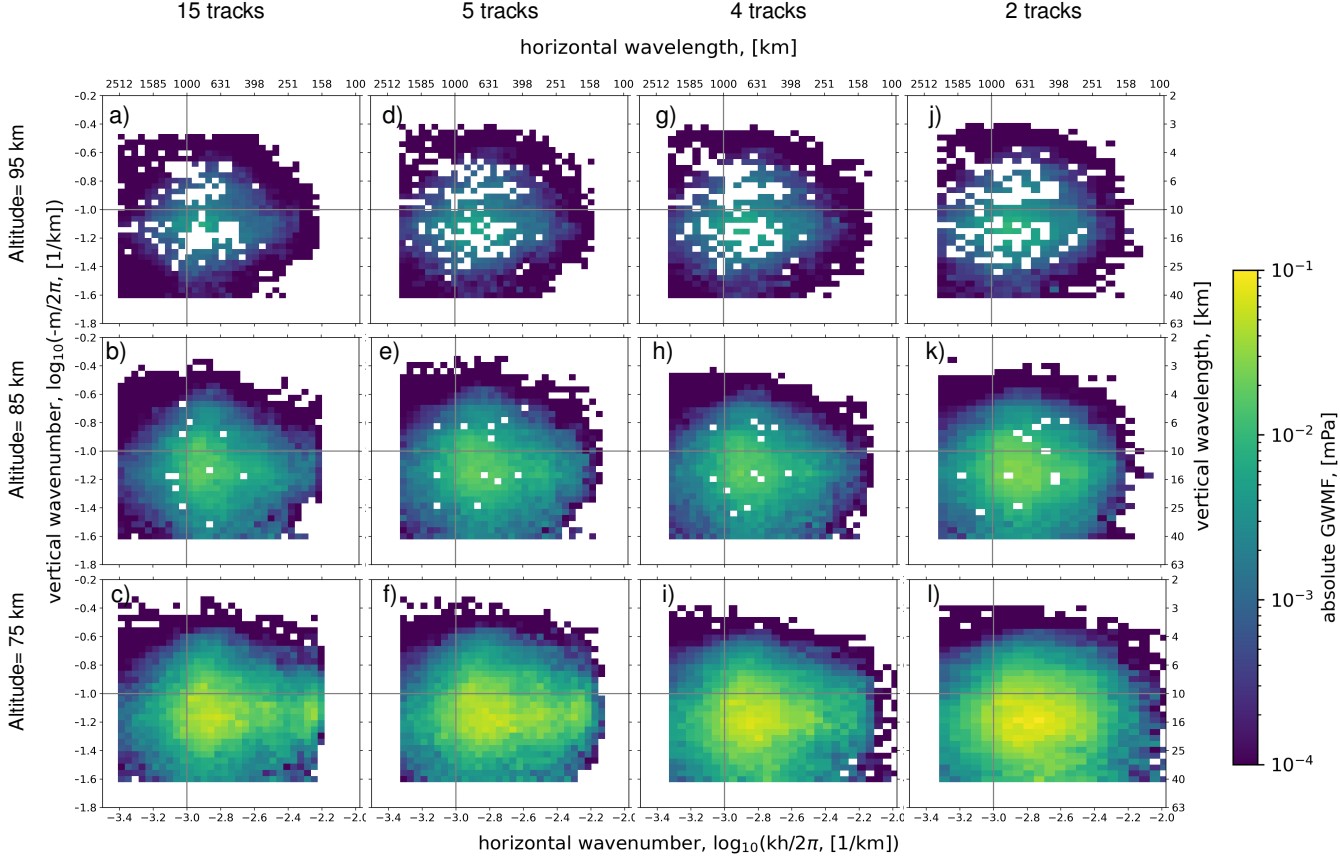

**Figure 10.** Distribution spectra of GW pseudomomentum flux versus logarithmic horizontal and vertical wavenumbers at 75 km (bottom panels), 85 km (middle panels) and 95 km (upper panels) altitude. Varying cube sizes in the across-track direction of 15 tracks (a-c), 5 tracks (d-f), 4 tracks (g-i) and 2 tracks (j-l) are applied. The 15 and 5 track analyses are for sampled data while 4 and 2 track analyses are for retrieval data with realistic noise. Flagged-out data due to insufficient fitting quality result in more blank bins for 95 km altitude. The gray reference lines through the plots indicate 1000 km horizontal and 10 km vertical wavelength, respectively.

the two satellites can be adjusted by pointing. We here assume a gap in the center which can be used to widen the total covered

region. In order to study the influence of the gap as well, we use an equidistant sampling of 5 points here, as well as a 4-track configuration in which the center track is not used. If only one SHI instrument would be operated, only two measurement tracks would be available. For this case, a series of S3D wave analyses were conducted with cube sizes of 5, 4 and 2 across-measurement-tracks on both sampled and retrieved HIAMCM temperature residual data. The 5-tack case is included in the simulation and discussed here as it serves as a bridge between the odd and even tracks, and offers an opportunity to examine

whether the S3D analysis can be performed normally on reduced number of tracks. Along-track and vertical size are kept the





same, i.e. 21 points and 15 points, respectively. The initial 15-track case is used as reference for inter-comparisons. We have evaluated all cases for both sampled and E2E data, but show exemplarily 15 tracks (Fig. 10 a-c) and 5 tracks (Fig. 10 d-f) for sampled data and 4 tracks (Fig. 10 g-i) and 2 tracks (Fig. 10 j-l) for full E2E data.

All three cases of reduced track numbers display the same major patterns as the reference case. It is noted that for the three cases, the spectra have a horizontal wavelength cutoff limited to around $2100\,\mathrm{km}$, a few hundreds kilometers shorter than the reference case of $2500\,\mathrm{km}$. Besides, the 4- and 2-track cases have a wider spread of wave distribution for an altitude of $75\,\mathrm{km}$, which reaches towards the short end of horizontal wavelength until $100\,\mathrm{km}$, while other cases are close to $150\,\mathrm{km}$. The two-track have highest PGWMF, but look somewhat blurred. In general, all track-combinations are suited to recover the spectrum.

### 4.2.2   Analysis of phase speed and wave direction


A more stringent test are spectra of PGWMF versus ground-based phase speed and direction. The direction is determined from the horizontal wave vector, the intrinsic phase speed can be calculated from the dispersion relation and the ground-based phase speed is then calculated via Doppler shifting the phase speed with the large-scale winds. The according PGWMF distribution is shown in Fig. 11 for latitudes of $40°\mathrm{N}$ to $70°\mathrm{N}$. The panels are shown for the same number of tracks and the same altitudes as

in Fig. 10. The four track-number cases show common characteristics on both the phase speed values and wave directions. At $75\,\mathrm{km}$ altitude there are two lobes, one towards north-northeast (NNE) and a second towards southwest (SW). Maximum phase speeds in these lobes are around $50\,\mathrm{m/s}$. In addition, there is a wide-spread background of waves propagating (ground-based) to the east. At $95\,\mathrm{km}$ altitude, all low phase speed waves are strongly attenuated, the north-northwest (NNW) lobe completely removed presumably by critical level filtering. The fast eastward waves now prevail. The 2-track data seem to have wider

spread and some additional features which may be misinterpretations of waves. This is more pronounced at 75 km at the lower edge of the emission layer, where the noise level is higher. In general, however, all cases reproduce the same salient features.

The deviation of wave direction caused by reducing the number of measurement tracks is further examined by scatter plots of wave directions for the various track-number cases (y-axis) against the 15-track reference (x-axis) in Fig. 12. The distribution is for the whole globe. We find two clusters of main propagation, a larger around $180°$ and a smaller around $0°$. For ideal fits,

we expect identity, i.e. the same leading waves with the same wave directions would be identified independent of the number of tracks and independent of imposed noise. In this case, all points would be on the 1:1 identity line. Indeed, we find that most points cluster around the identity line. There are interesting deviations, though. There are smaller clusters around $(0°, 180°)$, $(180°, 0°)$, $(360°, 180°)$ and $(180°, 360°)$, which indicate direction flips. The number of direction flips increases with fewer tracks. For the two track data there is a general loss of ability to determine the propagation direction, which is expressed in

vertical stripes around the preferred propagation direction centers. Again, the loss of direction information is most pronounced at 75 km altitude.





**Figure 11.** Polar plots of phase speed and direction versus GW pseudomomentum flux in the northern middle and higher latitudes of $40°$ to $70°$ at an altitude of $75\,\mathrm{km}$ (bottom panels), $85\,\mathrm{km}$ (middle panels) and $95\,\mathrm{km}$ (upper panels) in line with the four cases as Fig. 10. The direction is in azimuth angle by which $0°$ is eastward and $90°$ is northward. The white dashed radius lines indicate various phase speed values in unit of $\mathrm{m \cdot s^{-1}}$.

### 4.2.3 Analysis of zonal mean GW momentum flux

Zonal mean GWMF and its vertical gradient is the primary goal of the mission and the most stringent test we can apply in the assessment. The dynamical driving and hence the overall structure of the MLT is largely governed by large-scale wind acceleration due to GW dissipation. Studies of equatorial oscillations such as mesospheric semiannual oscillation (MSAO) and mesospheric quasi-biennial oscillation (MQBO), of the general mean circulation and the temperature structure as well as of the reappearance of the stratopause after a sudden stratospheric warming are largely based on zonal mean GW activity and would



**Figure 12.** Scatter plots of wave direction of the first wave component for 5 tracks (a-c), 4 tracks (d-f) and 2 tracks (g-i) on the y-axis versus the reference of 15 tracks on the x-axis at an altitude of $75\,\mathrm{km}$ (bottom panels), $85\,\mathrm{km}$ (middle panels) and $95\,\mathrm{km}$ (upper panels). A black 1:1 identity line is added for the comparison. Note that points around $(0°, 360°)$ and $(360°, 0°)$ appear simply due to a mapping of $360°$ on the y or x axis in plotting.

highly benefit from accurate PGWMF estimates. This is further explicated is Sect. 5. Furthermore, zonal-mean GWMF can be inferred as a true reference from the winds directly as introduced in Sect. 2. We here discuss the influence of the observation

method on this primary observation aim.

Fig. 13 a-d & f-i depict altitude-latitude cross-sections of the zonal average GWMF in $1°$ latitudinal bins for the four cases of different numbers of observation tracks. Both zonal and meridional component are given. The GWMF is inferred from temperature residual data via S3D wave analysis and polarization relations. We use one snap shot of HIAMCM model data



**Figure 13.** Zonal (a-d) and meridional (f-i) zonal mean GW momentum flux in 1° latitude bins from sampled and retrieved orbit-track HIAMCM temperature data, using S3D fitting method and polarization relations. Four cases as Fig. 10 are listed. GWMF values (e, j) directly inferred from wind fluctuations are also given, which are running averaged over 5° latitude bins and vertically running averaged over 5 km altitudes.

but sample with a series of orbits corresponding to one week of measurements. In this way we separate sampling issues from

general limitations of the method.

Global distribution of GWMF values directly inferred from wind fluctuations as Eq. 9 is illustrated in Fig. 13 e & j, which is regarded as a reference of our comparisons. The values are running averaged over 5° latitude bins and vertically running




averaged over 5 km altitudes. Detailed inter-comparison line plots of momentum flux values from the four cases to the values computed from wind residual data are presented in Fig. 14 & 15 & 16 for altitudes of 75 km, 85 km and 95 km, respectively.

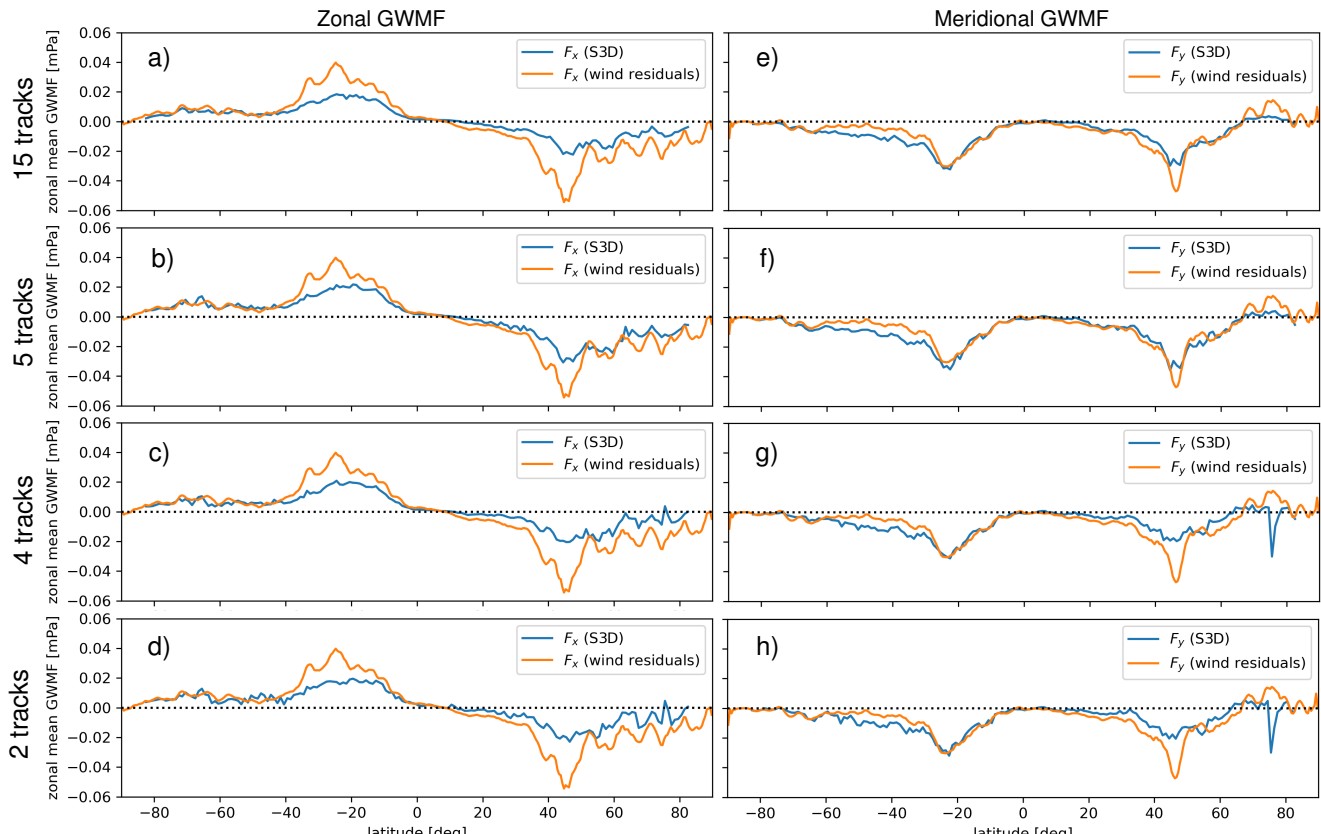

**Figure 14.** Line comparison plots of zonal (a-d) and meridional (e-h) zonal mean GW momentum flux calculations in $1°$ latitude bins from S3D analysis (blue line) and directly from wind fluctuations (orange line) for an altitude of 75 km. It is in line with the four cases as Fig. 10. GWMF from wind residuals are running averaged over $5°$ latitude bins and vertically running averaged over 5 km altitudes.

The main features of Fig. 13 show that GWs intrinsically propagate against the prevailing zonal wind, i.e. eastward propagation in the summer mid mesosphere around 70 km and westward propagation in the winter hemisphere. The MLT is characterized by strong wind gradients and according wave dissipation and critical level filtering. This is expressed by strong gradients of GWMF, that the absolute values of momentum flux change by two orders of magnitude, and partly, in the zonal direction a reversal of the direction of zonal GWMF. This zonal mean behaviour is consistent with the phase speed spectra of Fig. 11, which show the filtering of the high-GWMF but slow westward GWs between 75 km and 95 km altitude and, accordingly, prevailing fast eastward waves at 95 km altitude. This pattern is captured by all four cases of different track numbers.

    The corresponding line plots indicate a good agreement of the momentum flux values inferred from the observed temperature residuals with the reference from model wind fluctuations. Some deviations are found in the mid-latitudes for zonal GWMF





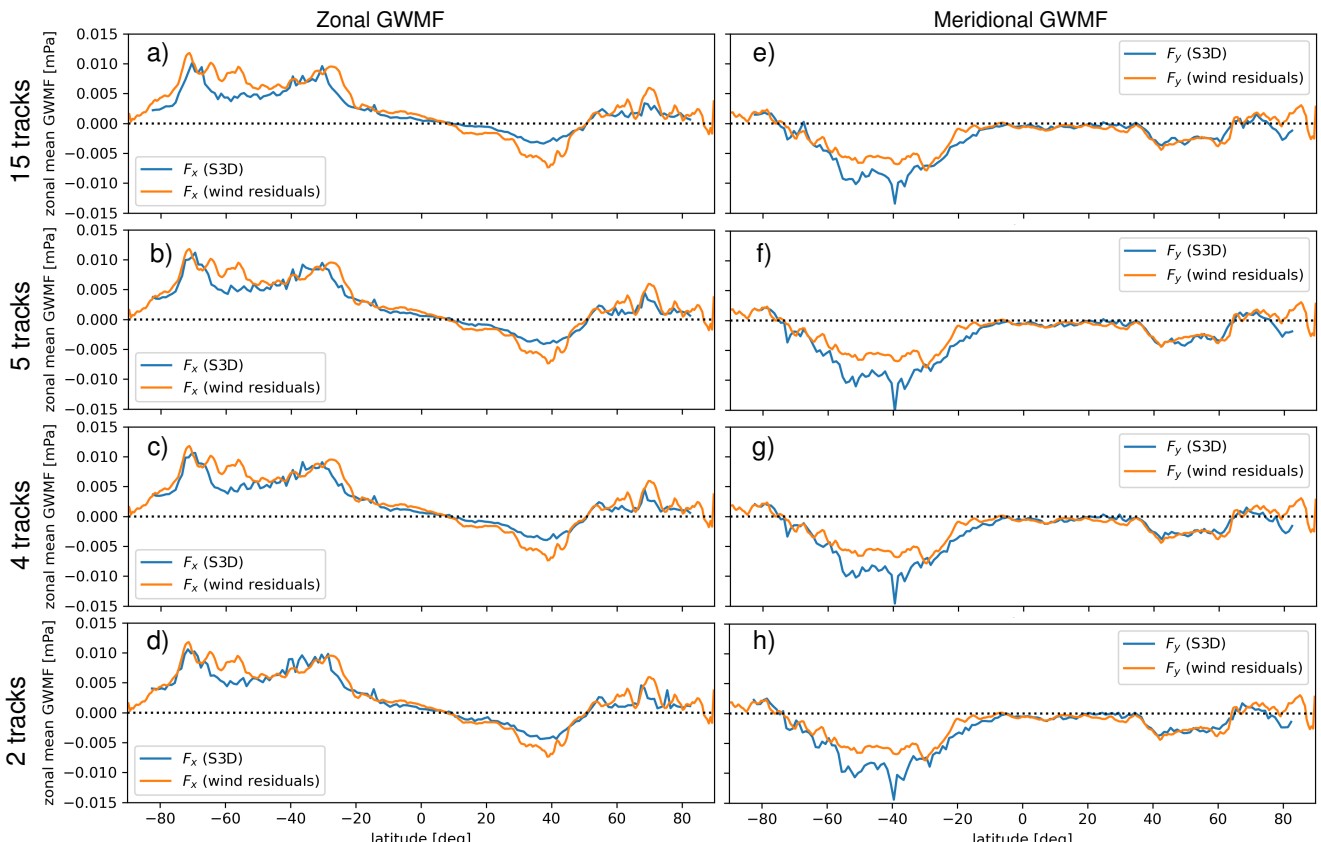

**Figure 15.** Same as for Fig. 14 but for an altitude of $85\,\mathrm{km}$.

components especially at an altitude of $75\,\mathrm{km}$, at $30°\mathrm{S}$ and $40°\mathrm{N}$. For meridional GWMF components, discrepancies mainly

appear in the southern hemisphere at $40°\mathrm{S}$-$60°\mathrm{S}$ for altitudes of $85\,\mathrm{km}$ and $95\,\mathrm{km}$. These differences are due to strong vertical gradients and could be either a problem of the wave fitting method to identify the local vertical wavelength associated with the cube center altitude or an effect of non-linearity and limitations to the linear GW physics employed to calculate GWMF.

In conclusion, the assessment results show that wave analysis on tomographic temperature observations of few observation tracks (down to 4 tracks and even 2 tracks) are suitable to gain reliable zonal means of zonal and meridional GWMF.

**4.3 How much noise can we afford?**

The E2E assessment performed in Sect. 4.2 indicates the viability of the proposed mission concept based on our best estimate of instrument performance. In order to investigate whether further miniaturization of the instrument and related decrease of the signal-to-noise ratio would be feasible, the best-estimate noise level superposed on the synthetic spectra is scaled by multiples of 2, 4, 6, 16, and 32. After that, temperature retrieval and S3D wave analysis are performed as before. We focus on the 4-track



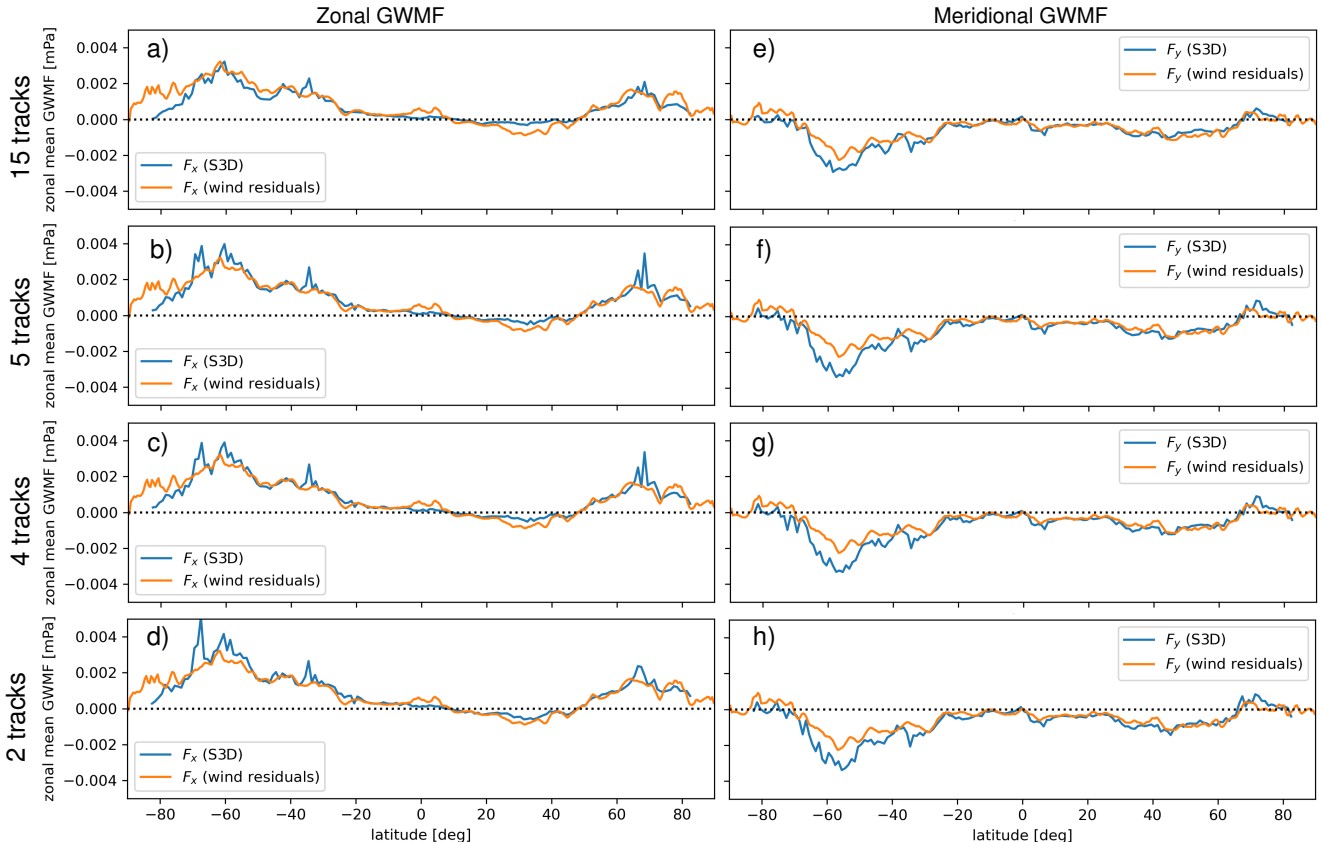

**Figure 16.** Same as for Fig. 14 but for an altitude of $95\,\mathrm{km}$.

data and assess the E2E results by zonal mean zonal GWMF values. We show in Fig. 17 color-coded line plots for noise levels of 1, 2, 4, 8, 16 and 32 for altitudes of 75, 85 and $95\,\mathrm{km}$, respectively. The resulting distributions displayed in Fig. 17 are more noisy and coarser since only a single day of orbits was used. The zonal mean GWMF plots indicate that the GW structure tends to be damped with increasing noise level, but the main features are retained up to a noise level of 8 times the original. For a 32-fold noise level, the wave signals are no longer discernible. Further detailed analysis of zonal mean GWMF for a 4-fold noise level is given in Appx. D. In general, our approach could tolerate enhanced noise up to a factor of 4 higher than our best estimate of the instrument of Kaufmann et al. (2018), which allows for further downsizing of the instrument or higher detector temperatures resulting in higher dark current levels and shot noise.

## 5    Discussion of science applications

Which scientific questions could be directly addressed, if an instrument such as discussed in this paper would be in orbit? And for which studies would we need further ancillary data? A two-step processing chain as it would be applied to in-orbit data is

**Figure 17.** Line plots of the zonal components of zonal mean GW momentum flux in 5° latitude bins from S3D analysis on retrieved temperature residuals with estimation of different noise levels of 1 (orange line), 2 (blue line), 4 (green line), 8 (cyan line), 16 (magenta line), 32 (yellow line) at an altitude of 75 km (a), 85 km (b) and 95 km (c). A 4-track case is applied for this analysis.



sketched in Figures 18 and 19. The first part relies only on data directly obtained by retrieval and wave analysis, the second would involve atmospheric winds generated e.g. by data assimilation.

Gravity wave momentum flux can be generated solely from observations made by the proposed instrument (Fig. 18). From the GWMF values of single GW events climatological distributions, such as average monthly mean maps, zonal means, drag
from vertical gradients of GWMF, and spectral distributions in terms of horizontal and vertical wavelength or intrinsic phase speed, can be generated. Such distributions are always the starting point of more in depth scientific investigations and can be used for interpretations of potential sources. Together with climatologies of winds and temperatures from other observations covering the MLT altitude range (e.g. the URAP climatology (Swinbank and Ortland, 2003)) they can be used to gain a first assessment of the momentum balance. Finally, spectral distributions including the direction can be used to distinguish different
pathways into the MLT (as outlined in the introduction) and hence provide the information to distinguish between conceptually different modeling approaches (both GW allowing GCM, which still contain tunable parameters, and explicit physical GW models) than the GW variances and absolute values of GWMF we can use nowadays.

Even more investigations are facilitated, if also estimates of the large scale winds at observation time are available. In the stratosphere such wind data are regularly generated by assimilation systems operated by numerical weather prediction centers.
In the mesosphere often geostrophic winds are used and zonal mean values were generated up to 90 km altitude (Ern et al., 2013; Smith et al., 2017). In order to gain a 3D and time dependent picture including tides, data assimilation is a powerful tool (e.g., Eckermann et al., 2009; Pedatella et al., 2018, 2020).

Zonal mean winds already allow to assess the driving of large scale wind patterns by GWs. Examples for these are studies based on absolute values of GWMF from limb scanning. These are the most reliable estimate of global GWMF distributions
we can currently gain, but they are limited because of the lack of direction information. In zones of strong vertical wind shear and an environment where slow phase speed GWs dominate, we can assume that the vertical gradient of the momentum flux corresponds to drag directed opposite to the shear (i.e. by GWs causing the shear layer to propagate downward). This concept has been very successfully used e.g. in studies of the stratospheric quasi-biennial oscillation (Ern et al., 2014), of the excitation of quasi-two-day waves (Ern et al., 2013) and of the role of GWs in sudden stratospheric warmings (Ern et al.,
2016). However, in regions where substantial filtering of slower phase speed waves has occurred already at lower altitudes and only faster phase speeds survive or when GWs dissipate primarily due to the increasing amplitudes the waves attain when they propagate upwards into regions of lower density, such simplifications cease to work. That can be seen for instance in the discussion of the mesospheric semiannual oscillation (Ern et al., 2015, 2021) where arguments for the direction of drag became complex and indirect. This increased complexity is mark of many of the large-scale global wind patterns in the mesosphere.
The proposed instrument combined with background wind information would resolve ambiguities such as encountered by Ern et al. (2015, 2021) immediately by combining the observed GW parameters with estimates of background winds (cf. Fig. 19. For instance, the ground-based phase speed spectrum could directly be compared to the wind velocities and the type of GW dissipation (critical level versus saturation of growing amplitudes) inferred. Blocking diagrams as introduced by Taylor et al. (1993) could be used to understand the actual interaction of GWs and background wind. The vertical gradient of the
zonal or regional mean of GWMF can be used to infer net drag including its direction and taking into account GWs from all





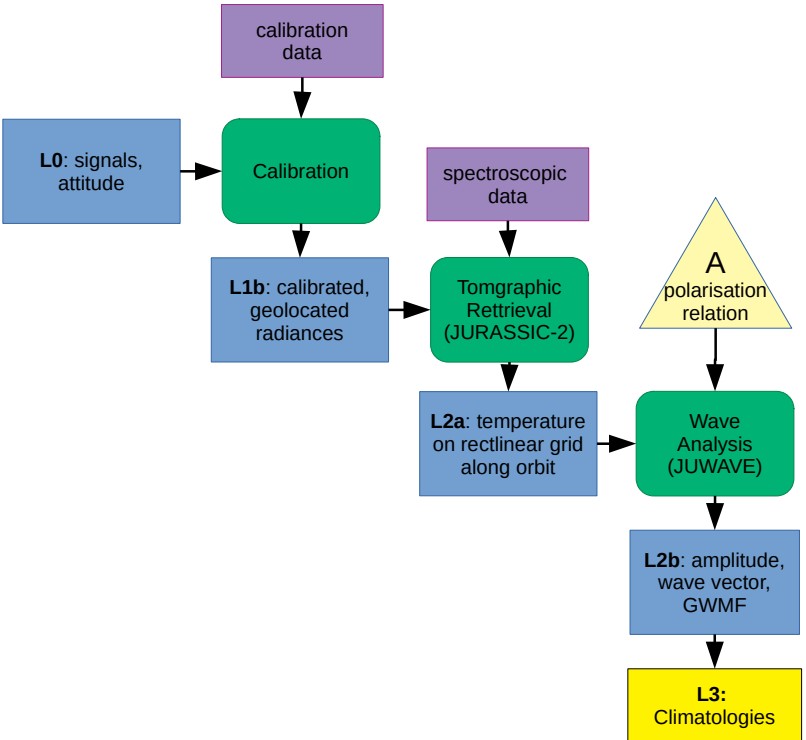

**Figure 18.** Processing chain including retrieval and GW analysis. Up to a climatology of GWMF in terms of e.g. zonal means or monthly mean 3D global distributions (Level 3 data) only data observed by the instrument are required.

propagation directions. Also in complex situations such as the mesospheric SAO the contribution of GWs to the driving can be directly estimated. For such applications zonal mean winds or coarse spatial representations would suffice and a general guidance of the GCM by the assimilation system would produce novel insights.

A more sophisticated way to study the interaction of GWs and large-scale winds is forward and backward ray-tracing. This is more accurate than blocking diagrams as for instance lateral propagation and avoidance of critical levels is taken into account (Preusse et al., 2009a; Kalisch et al., 2014; Thurairajah et al., 2014; Chen et al., 2019; Thurairajah et al., 2020). Furthermore, tides cause changes of the large scale winds at similar time scales as the periods of the GWs propagating through these winds. This causes refraction of the ground-based wave frequency (Senf and Achatz, 2011) thus also allowing waves, which would be expected to be filtered by a tide based on its original ground based frequency at source level, to propagate further. Having

identified critical levels in backward ray-tracing, however, this is a clear sign for in situ generation of GWs in the middle atmosphere by unstable jets or secondary wave generation. Also for such studies a coarser representation of the background winds is sufficient.

    Most demanding is the backward tracing of observed waves to potential tropospheric sources. Backward or forward ray-tracing of individual wave events requires a full wave characterisation and for temperature observations requires hence three





dimensions. Backward ray-tracing from 3D data was used in previous studies for source identification. Examples for mesoscale stratospheric waves are studies by Preusse et al. (2014); Krisch et al. (2017); Perrett et al. (2021); Strube et al. (2021); Geldenhuys et al. (2021). Also high frequency waves in the MLT have been studied by such means (Wrasse et al., 2006; Pramitha et al., 2015). Backward ray-tracing is a safe method to infer whether GWs can be followed down to the troposphere or meet an intransparent level higher up - the latter indicates a source in the middle atmosphere for instance by secondary wave gen-

eration. This is an advance over simply considering phase-speeds at the observation and wind fields below as all the spatial and temporal changes of the wave parameters along the ray-path are taken into account. Whether the precision of the wave parameters and the accuracy of the background winds from assimilation will be sufficient to trace mesoscale waves back to individual tropospheric sources will need to be studied in more detail. Upward ray-tracing then can be used to investigate the interaction of GWs with the background and to provide a complementary drag assessment to the vertical gradient of PGWMF.

For instance it could be assessed which waves can reach the MLT.

In the context of momentum balance studies, the observational filter of a limb sounder needs to be taken into account. The observational filter allows to observe only waves with horizontal wavelengths longer than O($100\,\mathrm{km}$). It is evident that this misses an important part of the GW spectrum. However, high-altitude GCMs without GW parametrization are even more restricted to long horizontal wavelengths than limb sounding observations (e.g. Sato et al., 1999, 2012; Siskind, 2014; Becker

and Vadas, 2020). It is one of the puzzles of atmospheric dynamics that these GCMs still produce a realistic atmosphere even above the stratopause where GWs become increasingly important. For the stratosphere nowadays global simulations with a few kilometer and even down to $1\,\mathrm{km}$ grid distance are available (e.g. Stephan et al., 2019a, b; Polichtchouk et al., 2022) which allow to assess how large a fraction of GWMF is likely to be missed due to the observational filter. However, such simulations have an upper altitude limit of less than $80\,\mathrm{km}$ and most of them damp GWs in a strong sponge layer above

$40\,\mathrm{km}$ (Preusse et al., 2014, and references therein). The best estimate for such lower altitudes is that GWMF is about equally partitioned to scales shorter and longer than $100\,\mathrm{km}$ horizontal wavelength, respectively. The almost vanishing influence of the observational filter in our study is thus due to the fact that current state of the art models do not resolve the shorter scales and it is reasonable to assume that in reality only roughly half of the GWMF would be observed also in the mesosphere. The partitioning of momentum flux between different wavelength regions is, however, one of the major unknowns in the field

of GWs and low-error, well-characterized global observations of GWMF would be an essential and necessary step forward to answer this fundamental question. A brief outline of the current state of knowledge is given in appendix E. In the lower thermosphere short period, short horizontal wavelength GWs become strongly damped (e.g., Pitteway and Hines, 1963; Vadas, 2007; Yigit et al., 2009) and hence likely an even larger fraction of that part of the GW spectrum for which GWs can propagate upward is quantified.

**6   Conclusions**

The energy and momentum gravity waves deposit in the mesosphere and lower thermosphere is a major driver of the dynamics and hence the entire structure of this region. In order to understand these processes, we require global observations of gravity





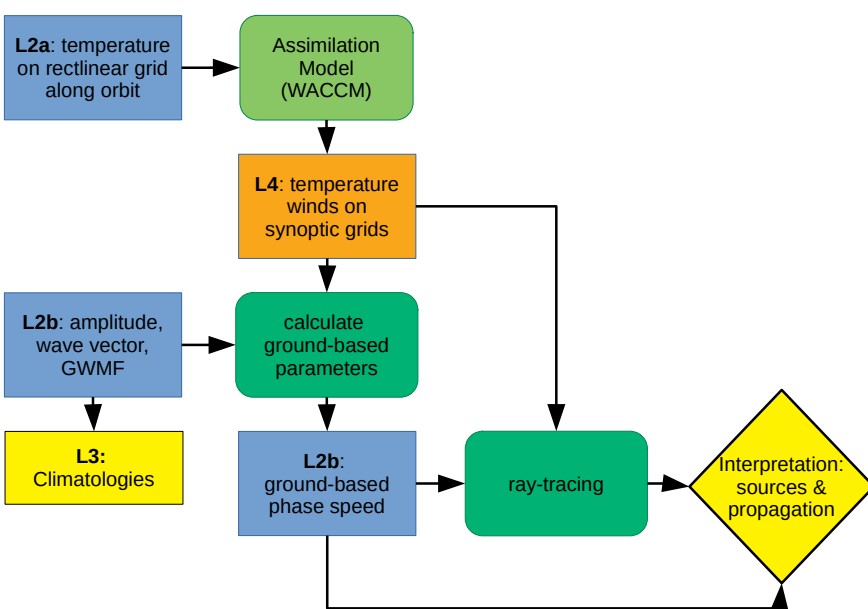

**Figure 19.** Processing chain for scientific interpretation of propagation, source identification and critical level filtering. In order to determine the ground-based frequency and perform ray-tracing background wind velocities for the observation location and time are required. These can be generated as Level 4 data via assimilating various data sets (including the here described observations) in a consistent manner.

wave pseudomomentum flux (PGWMF), its vertical gradient and its spectral composition. In particular, the distribution of PGWMF with respect to phase speed and direction allows direct inference of the interaction of the GWs with the background
winds and the mechanisms of how they drive global scale patterns such as the general mean circulation and tropical oscillations. By observing 3D temperature structures, one should be able to gain such information from space. The current study investigates aspects of the concrete realizations and whether it is sufficient to sample gravity waves in that part of the atmosphere with only 2–4 instantaneous field-of-views.

To assess this question, an end-to-end simulation comprising orbit simulation, temperature retrieval and 3-D wave analysis
was performed and the performance was assessed on the basis of GWMF comparisons. We considered zonal mean values of GWMF deduced from the simulated observations and compared this to a reference distribution deduced directly from the model winds. In addition, we investigated phase speed spectra and the ability to characterize the direction of single wave events.

Our simulations show that zonal mean values are most robust and that the direction of wave events is most sensitive with respect to both noise and a low number of observation tracks. In general, a four track instrument with nominal performance
provides reliable wave quantification and can be employed both for the generation of zonal mean climatologies as well as





studies of the interaction of GWs with the background winds. The full wave characterization of the 3D data will facilitate ray-tracing studies, which provide a powerful tool for the discussion of potential sources and wave-meanflow interaction.

For reasons of computational efficiency, we made the following assumptions: Since the HIAMCM model resolves only GWs longer than about $200\,\mathrm{km}$ horizontal wavelength, we have not included the scale separation in the assessment process

and we assume a constant track distance of the split-interferogram of 30 km (which implicitly assumes a constantly lit scene). Such assumptions are necessary to make the study feasible and are, on the whole, justified, but they tend to overestimate the reliability of the results somewhat. An important finding is that the signal-to-noise levels of the temperature data assumed in this study can be relaxed by a factor of four to allow for further miniaturization of the observation system. For the gravity wave spectrum used in this study, two field-of-views parallel to the orbital track are sufficient to resolve most of the spectral

distribution, whereas the direction of individual GWs cannot be resolved as well as in simulations assuming four or more tracks.

Summarizing, the most important conclusions obtained are listed below:

– Assumptions made for the S3D fitting technique are non-critical in practical application.

– Polarization relations are valid up to $130\,\mathrm{km}$ altitude.

– Four closely spaced field-of-views are sufficient to resolve the spectral distribution and the direction of gravity waves in the mesosphere. Two field-of-views are sufficient to resolve the spectral distribution, but uncertainties of inferred GW direction information are increased.

– Inferred distributions are still meaningful even if noise is increased by a factor of four with respect to the assumed instrument performance.

**Appendix A: Estimation of the large-scale background from MLT satellite observations**

One of the first steps in any GW analysis using satellite data is the separation of observations into a large-scale atmospheric background and small-scale fluctuations that can be attributed to GWs. Usually, this is performed via scale separation, either vertically or horizontally.

Compared to GWs, planetary waves (PWs) in the extratropics often have very long vertical wavelengths. Therefore, one

method to separate the GW signal is vertical high-pass filtering of observed altitude profiles. This method is often applied to observations from single ground based stations (e.g., Ehard et al., 2015; Strelnikova et al., 2021), but also to satellite data (e.g., Tsuda et al., 2000; de la Torre et al., 2006; Gavrilov, 2007; Rapp et al., 2018a). One drawback of this method is that with high-pass filtering a large part of the GW vertical wavenumber spectrum is also lost. Another drawback is the existence of short-vertical-wavelength global-scale wave modes, for which a separation from GWs by vertical filtering is not possible. Such

global-scale wave modes are, for example, inertial instabilities (Rapp et al., 2018b; Strube et al., 2020), or tropical wave modes like Kelvin waves (Ern et al., 2008, 2009; Ern and Preusse, 2009). Consequently, GW fluctuations resulting from vertical high-



pass filtering can be strongly biased. Another problem in the MLT region as compared to the troposphere and stratosphere is that, on average, the spectrum of GW vertical wavelengths is shifted to longer wavelengths.

For this reason, several approaches of separation of horizontal scales were developed. One approach is horizontal high-
pass filtering by local averages, or local fits. This approach comprises, at a fixed altitude, high-pass filtering into the direction along the satellite measuring track (e.g., Yan et al., 2010; Trinh et al., 2018), high-pass filtering in across-track direction (e.g., Alexander and Barnet, 2007; Ern et al., 2017), or a combination of both. These easy-to-implement methods provide reliable results, but require dense sampling. Furthermore, a large part of the GW horizontal wavenumber spectrum is lost by horizontal high-pass filtering.

This is why several methods that utilize the global coverage provided by satellite data to directly fit global-scale waves were developed. Some approaches estimate stationary waves using dedicated methods (e.g., Alexander et al., 2008b; Wright et al., 2010), or stationary and slowly traveling waves using Kalman filtering (e.g., Ern et al., 2004).

These approaches are not well suited for the mesosphere, because PWs in the mesosphere, like 2-day waves, or ultra-fast Kelvin waves (e.g., Garcia et al., 2005; Ern et al., 2009, 2013), can travel quite fast. In addition, at high altitudes tides
become increasingly important (e.g., Mukhtarov et al., 2009; Pancheva et al., 2009, 2010). To overcome these difficulties and to preserve a portion of the observed GW spectrum as large as possible, a dedicated method has been developed by Ern et al. (2011, 2013, 2018). This method has been applied successfully already in several studies in the MLT using SABER data, and it performed very well (e.g., Ern et al., 2011, 2013, 2016, 2021). Furthermore, this made it possible to create a global climatology of GWs that extends up to $90\,\mathrm{km}$ altitude (Ern et al., 2018).

In a first step, for a set of fixed latitudes and altitudes, 2D spectra in longitude and time are calculated in overlapping time windows of $31\,\mathrm{days}$. According to the orbit geometry of satellites in low Earth orbit, zonal wavenumbers up to about 7 can be resolved by the satellite sampling, and wave frequencies up to $\sim$1 cycle/day if both ascending (satellite flying northward) and descending (satellite flying southward) orbit parts are combined (Salby, 1982). If only one orbit leg is available (either ascending, or descending), wave frequencies up to $\sim$0.5 cycles/day can still be resolved. Small-scale GWs are not resolved
by the satellite sampling and appear as uniform white noise in these spectra. The 2D zonal wavenumber - frequency spectra are then back-transformed into physical space for the largest spectral components (these can be attributed to PWs) for the exact time of each observation, and interpolated spatially to its exact location. Together with the zonal average, these estimates of large-scale waves are then subtracted from each observation. In this first step, atmospheric solar tides are not considered because of their amplitude variation on time scales shorter than $31\,\mathrm{days}$.

Tides, however, can easily be removed in a second step. This is facilitated by the orbit geometry of satellites in low Earth orbits. LEO satellites cross a given latitude at two local solar times (LSTs), one corresponding to the ascending orbit leg, the other corresponding to the descending orbit leg. As for most LEO satellites the orbits are only slowly precessing, these two local times are about fixed on the time scale of a few days. This has the effect that, if only observations of one orbit leg (either ascending, or descending) are considered, solar tides appear as a stationary wave pattern, or just as an offset, in the
data. If ascending and descending orbit data are treated separately, these stationary patterns and offsets can easily be fitted and





subtracted from the fluctuations obtained after the first step. This procedure can even be performed if only data of one of the orbit legs is available, and a dedicated analysis of tidal modes is not required (e.g., Trinh et al., 2018).

The resulting fluctuations can then be attributed to GWs, with the advantage of not cutting down the GW spectrum by high-pass filtering, as was the case for the other methods. Still, additional filtering can be performed to adapt the scales of GWs
contained in the data to any GW analysis method that one would like to apply afterwards.

**Appendix B:  Photochemical reaction parameters used in the $O_2$ airglow emission simulation**

Following Sheese (2009), the A-band volume emission rate is given by

$$\eta = (P_{\text{A−band}} + P_{\text{B−band}} + P_{\text{Phot}_{O_3}} + P_{\text{Phot}_{O_2}} + P_{\text{Barth}})\frac{A_{762}}{P_{\text{loss}}}, \tag{B1}$$

where the first term describes the production rate of the excited state $O_2(b^1\Sigma_g^+, v=0)$ and the second term is the fraction of
the $O_2$ A-band compared to all losses of the excited state $O_2(b^1\Sigma_g^+, v=0)$. $A_{762}$ is the Einstein-coefficient of the A-band and can be calculated by $A_{762} = F_c A_{1\Sigma}$ where $F_c$ is the A-band Franck-Condon factor and $A_{1\Sigma}$ is the Einstein coefficient of all radiative transitions from the excited state $O_2(b^1\Sigma_g^+, v=0)$ corresponding to reaction 19 in Table B1. In the following, $[\cdot]$ correspond to number densities of molecules. The loss component can be calculated by

$$P_{\text{loss}} = A_{1\Sigma} + k_0[N_2] + k_3[O_3] + k_4[O_2] + k_6[O], \tag{B2}$$

where $k_0$, $k_3$, $k_4$ and $k_6$ correspond to the quenching rates of the reactions 20–23 in Table B1. The A-band photo-absorption is calculated by

$$P_{\text{A−band}} = g_A[O_2], \tag{B3}$$

where $g_A$ is the photochemical reaction coefficient of the reaction 1 in Table B1. The B-band photo-absorption is calculated by

$$P_{\text{B−band}} = \frac{Kg_A[O_2]}{A_{771} + K + K_{3B}[O_3]} \tag{B4}$$

$$K = K_{0B}[O] + K_{1B}[O_2] + K_{2B}[N_2], \tag{B5}$$

where $K_{0B}$, $K_{1B}$, $K_{2B}$ and $K_{3B}$ are the quenching rates of reactions 4–7, $g_B$ is the photochemical reaction coefficient of the reaction 2 and $A_{771}$ is the Einstein coefficient of the reaction 3 in Table B1.

The production rate due to the photolysis of $O_2$ and $O_3$ can be calculated by

$$P_{\text{Phot}_{O_2}} = \frac{(J_{\text{SCR}} + J_{L_\alpha})[O_2]\varphi k_1[O_2]}{A_{1D} + k_1[O_2] + k_2[N_2]} \tag{B6}$$

$$P_{\text{Phot}_{O_3}} = \frac{J_H[O_3]\varphi k_1[O_2]}{A_{1D} + k_1[O_2] + k_2[N_2]} \tag{B7}$$



where $J_{\text{SCR}}$, $J_{L_\alpha}$ and $J_H$ are the photolysis coefficients of the reaction 8, 9 and 10, $A_{1D}$ is the Einstein-coefficient of the reaction 11 and $k_1$ and $k_2$ are the quenching rates of reactions 13 and 14 in Table B1. $J_H$ is given by Sheese (2009). $J_{\text{SCR}}$ and $J_{L_\alpha}$ are calculated by following Sheese (2009). Hereby, the Lyman-$\alpha$ absorption cross-section and absorption quantum yield are set to $1.0 \times 10^{-20}\,\text{cm}^2$ and $0.55$, respectively (Reddmann and Uhl (2003)). Regarding the Schumann-Runge continuum, the absorption cross-section is given by Yoshino et al. (2005) over the given wavelength range and the absorption quantum yield is set to 1 (Brasseur and Solomon (2005)). The solar flux at the top of the atmosphere is taken from the SORCE Solar Spectral Irradiance (SSI) and can be obtained from https://lasp.colorado.edu/home/sorce/data/.

The production rate due to the Barth process can be calculated by

$$P_{Barth} = \frac{k_5 [O]^2 [M][O_2]}{C_{O_2}[O_2] + C_O[O]}, \tag{B8}$$

where $k_5$ is the quenching rate of reaction 15 in Table B1, and $C_{O_2}$ and $C_O$ are the fitting parameters of the simplified model proposed by McDade et al. (1986). $C_{O_2}$ describe the relative quenching rates of $O_2^*$ with $O_2$ corresponding to reaction 16 and $C_O$ describe the relative quenching rates of $O_2^*$ with O corresponding to reaction 17 in Table B1, which is a three step reaction as depicted in Fig. 4a.

## Appendix C: Known limitations of the O$_2$A-band forward model

To allow for repeated tomographic retrievals of a week of data, the employed forward model makes some simplifying assumptions that need to be revisited before applying it to actual measurements. First, for daytime measurements, the model requires the amount of in-scattered solar light at all positions. This is currently being tabulated based on climatological conditions and not based on actual volume-mixing ratios. Second, the model does not take the finite extent of the line shape into account. The support of the line-shape is very small compared to the spectral resolution of the instrument, such that this becomes problematic only for the treatment of self-absorption: we simplify this by computing the amount of self-absorption from the center of the line, which is a strict over-estimation of actual self-absorption that decreases in the line-wings. In this fashion, the access to radiation from lower altitudes (below $80\,\text{km}$) becomes increasingly limited, which we deem a worst case assumption for the purpose of this study, which is asserted by runs without any self-absorption that give better quality results down to $60\,\text{km}$ altitude. Both issues are straightforward to address by performing the required computations.

## Appendix D: Zonal mean distribution of GWMF for a 4-fold increased noise level

Illustrated in Fig. D1 is the global cross section distribution of zonal mean GWMF in $1°$ latitude bins, similar to Fig. 13c of the 4-track case, but for a noise level enhanced by a factor of 4. The general GW features in Fig. D1 are comparable to Fig. 13c & h, with the wind reversal and strong vertical gradients still clearly visible. In the altitude region above $80\,\text{km}$, these GW structures are in general well preserved in the averaged GWMF distribution, except for some outliers seen in the meridional components at $20°\text{N}$-$30°\text{N}$ for an altitude of around $85\,\text{km}$. Below $80\,\text{km}$, the latitude bins appear to be biased by noise perturbations.



**Table B1.** Photochemical reaction parameters used in the simulation of the A-band production and loss mechanisms

| Index | Reaction | Rate | Value | Unit | Reference |
|---|---|---|---|---|---|
| 1 | $O_2(X^3\Sigma_g^-) + hv(\lambda = 762.7nm) \rightarrow O_2(b^1\Sigma_g^+, v=0)$ | $g_A$ | $5.94 \times 10^{-9}$ | $s^{-1}$ | Bucholtz et al. (1986) |
| 2 | $O_2(X^3\Sigma_g^-) + hv(\lambda = 689.6nm) \rightarrow O_2(b^1\Sigma_g^+, v=1)$ | $g_B$ | $3.54 \times 10^{-10}$ | $s^{-1}$ | Bucholtz et al. (1986) |
| 3 | $O_2(b^1\Sigma_g^+, v=1) \rightarrow O_2(X^3\Sigma_g^-) + hv(\lambda = 771nm)$ | $A_{771}$ | $0.07$ | $s^{-1}$ | Rothman et al. (2013) |
| 4 | $O_2(b^1\Sigma_g^+, v=1) + O \rightarrow O_2(b^1\Sigma_g^+, v=0) + 0$ | $k_{0B}$ | $4.5e-12$ | $cm^3 s^{-1}$ | Yankovsky and Manuilova (2006) |
| 5 | $O_2(b^1\Sigma_g^+, v=1) + O_2 \rightarrow O_2(b^1\Sigma_g^+, v=0) + O_2$ | $k_{1B}$ | $4.2 \times 10^{-11} e^{\frac{-312}{T}}$ | $cm^3 s^{-1}$ | Yankovsky and Manuilova (2006) |
| 6 | $O_2(b^1\Sigma_g^+, v=1) + N_2 \rightarrow O_2(b^1\Sigma_g^+, v=0) + N_2$ | $k_{2B}$ | $5.0 \times 10^{-13}$ | $cm^3 s^{-1}$ | Yankovsky and Manuilova (2006) |
| 7 | $O_2(b^1\Sigma_g^+, v=1) + O_3 \rightarrow 2O_2 + O$ | $k_{3B}$ | $3.0 \times 10^{-10}$ | $cm^3 s^{-1}$ | Yankovsky and Manuilova (2006) |
| 8 | $O_2(X^3\Sigma_g^-) + hv(137nm \leq \lambda \leq 175nm) \rightarrow O(^3P) + O(^1D)$ | $J_{SCR}$ | see Sec. B | $s^{-1}$ | Sheese (2009) |
| 9 | $O_2(X^3\Sigma_g^-) + hv(\lambda = 121.6nm) \rightarrow O(^3P) + O(^1D)$ | $J_{L_\alpha}$ | see Sec. B | $s^{-1}$ | Sheese (2009) |
| 10 | $O_3 + hv(\lambda \leq 310nm) \rightarrow O(^1D) + O_2(a^1\Delta_g)$ | $J_H$ | $7.1 \times 10^{-3}$ | $s^{-1}$ | Sheese (2009) |
| 11 | $O(^1D) \rightarrow O + hv(\lambda = 630nm)$ | $A_{1D}$ | $6.81 \times 10^{-3}$ | $s^{-1}$ | Rothman et al. (2013) |
| 12 | $O(^1D) + O_2 \rightarrow O + O_2(b^1\Sigma_g^+, v=0)$ | $\varphi k_1$ | 0.95, see row 13 | unitless | Green et al. (2000) |
| 13 | $O(^1D) + O_2 \rightarrow O + O_2$ | $k_1$ | $3.3 \times 10^{-11} e^{\frac{55}{T}}$ | $cm^3 s^{-1}$ | Sander et al. (2011) |
| 14 | $O(^1D) + N_2 \rightarrow O + N_2$ | $k_2$ | $2.15 \times 10^{-11} e^{\frac{110}{T}}$ | $cm^3 s^{-1}$ | Sander et al. (2011) |
| 15 | $O + O + M \rightarrow O_2^* + M \ (M = N_2, O_2)$ | $k_5$ | $4.7 \times 10^{-33} \left(\frac{300}{T}\right)^2$ | $cm^6 s^{-1}$ | McDade et al. (1986) |
| 16 | $O_2^* + O_2 \rightarrow O_2(b^1\Sigma_g^+, v=0) + O_2$ | $C_{O2}$ | 7.5 | unitless | McDade et al. (1986) |
| 17 | $O_2^* + O \rightarrow O_2(b^1\Sigma_g^+, v=0) + O$ | $C_O$ | 33 | unitless | McDade et al. (1986) |
| 18 | $O_2(b^1\Sigma_g^+, v=0) \rightarrow O_2(X^3\Sigma_g^-, v=0) + hv(\lambda = 762.7nm)$ | $F_c A_{1\Sigma}$ | 0.93, see row 19 | $s^{-1}$ | Nicholls (1965) |
| 19 | $O_2(b^1\Sigma_g^+, v=0) \rightarrow products$ | $A_{1\Sigma}$ | 0.0878 | $s^{-1}$ | Rothman et al. (2013) |
| 20 | $O_2(b^1\Sigma_g^+, v=0) + N_2 \rightarrow products$ | $k_0$ | $1.8 \times 10^{-15} e^{\frac{45}{T}}$ | $cm^3 s^{-1}$ | Sander et al. (2011) |
| 21 | $O_2(b^1\Sigma_g^+, v=0) + O_3 \rightarrow products$ | $k_3$ | $3.5 \times 10^{-11} e^{\frac{135}{T}}$ | $cm^3 s^{-1}$ | Sander et al. (2011) |
| 22 | $O_2(b^1\Sigma_g^+, v=0) + O_2 \rightarrow products$ | $k_4$ | $3.9 \times 10^{-17}$ | $cm^3 s^{-1}$ | Sander et al. (2011) |
| 23 | $O_2(b^1\Sigma_g^+, v=0) + O \rightarrow products$ | $k_6$ | $8.0 \times 10^{-14}$ | $cm^3 s^{-1}$ | Sander et al. (2011) |

Particularly at 0°-30°N and 60°S-40°S no consistent GW patterns can be identified. It reveals that the wave analysis method becomes more sensitive to noises at lower altitudes. Overall, this GWMF distribution demonstrates that our approach of wave analysis is fairly stable despite of increasing the noise level by a factor of 4.

**Appendix E: The relative importance of different horizontal scales**

Coupling of the different layers of the atmosphere is achieved by freely propagating (i.e., internal) GWs, other wave modes such as evanescent waves or waves in a wave guide do not contribute. This limits the horizontal range of GWs which have this potential to horizontal wavelength $\lambda_h$ in the range of 20-30km on the short wavelength side to roughly 1000-2000km on the long wavelength side (Preusse et al., 2008). Shorter wavelengths exist as well (Fritts et al., 2017), but they are important in the GW cascade when it comes to transferring momentum and energy finally to turbulence, i.e., as the last step of the cascade. A

very rough, first approximation of the partitioning of GWMF between different scales can be gained by assuming a universal scaling law $T'^2 \propto k^{-5/3}$, where the wind or temperature fluctuations scale exponentially with the horizontal wavenumber. This





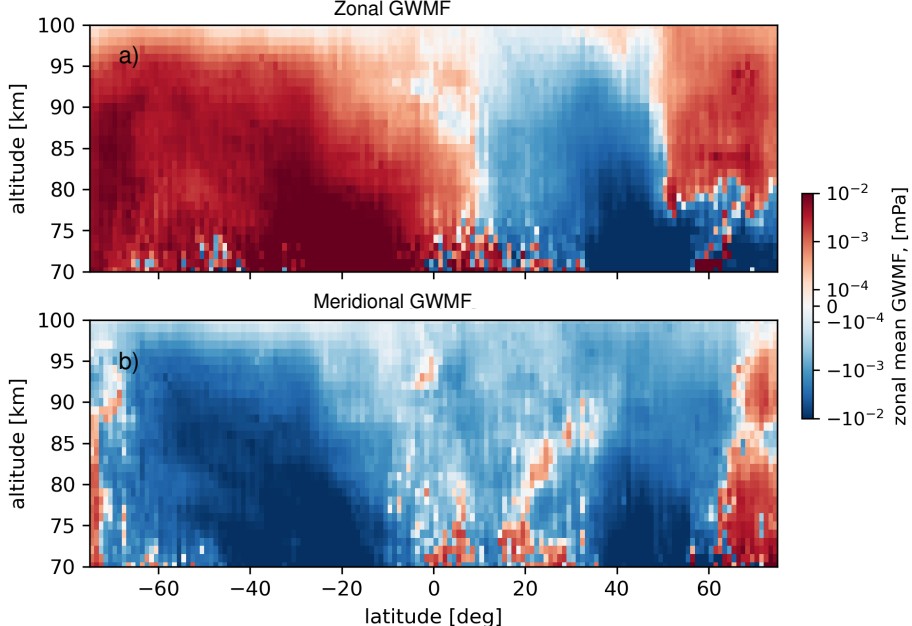

**Figure D1.** Zonal (a) and meridional (b) components of zonal mean GW momentum flux in $1°$ latitude bins from retrieved temperature residuals with an amplified 4-fold noise level applied to a 4-track case, using the S3D fitting method and polarization relations.

lets us expect that a horizontal wavelength of 100km roughly partitions GWMF into equal parts. A detailed review by Preusse et al. (2008) found that the available evidence at the time was roughly compliant with such a partitioning. Note that here we discuss the global scale: over localized sources shorter horizontal wavelengths sometimes have by factors larger GWMF (e.g. Kruse et al., 2022), but this is then connected also to a limited area. Which evidence did we gain since the previous review of Preusse et al. (2008)? First, there are now a number of studies using the vertical derivative of absolute GWMF inferred from limb sounding satellite instruments. It is argued in these studies that in regions of strong wind shear in particular those GWs break which are propagating opposite to the winds in the lower altitudes. Despite the fact that the absolute GWMF should be underestimated by the observation technique, the contribution to the total GW acceleration expected from momentum closure is very often more than 50% (Ern et al., 2013, 2014, 2015, 2016). Second, we now have more model studies with GW-permitting resolution not using any GW parametrization (e.g. Liu et al., 2014; Siskind, 2014; Becker and Vadas, 2020; Okui et al., 2021; Becker et al., 2022; Okui et al., 2022). These models reach at least similar performance for the MLT and better performance for the thermosphere than GW-parametrizing GCMs as they include middle atmosphere sources and oblique propagation, despite the fact that they resolve only GWs with horizontal wavelengths longer than $\approx$200km. Likely these models then somewhat overestimate the scales they resolve. Third, there are now GCM runs from NWP models with a much higher resolution (Stephan et al., 2019a; Polichtchouk et al., 2022). These data also indicate a roughly equal partitioning of GWMF between scales smaller and larger than a horizontal wavelength of $\lambda_h =$100km. Thus, assuming that horizontal wavelength larger than 100km convey



roughly half of the momentum flux is currently the best estimate we have. Unless we perform high-accuracy global GWMF observations this fundamental question will remain at least partly be subject to speculation.

*Author contributions.*   QC performed the GW calculation and track/noise-related feasibility assessment. KN was responsible for the forward model and interferogram split. BL performed the polarization relations validation. LK contributed with his knowledge about tomographic retrieval. ME was responsible the scale separation. PP performed the orbit simulation, prepared the JUWAVE S3D software and contributed with his expertise in GWs. PP and ME were responsible for the scientific application discussions. JU performed the tomographic temperature retrieval. EB provided the HIAMCM model data and contributed to the introduction. PP and MK initiated the topic and supervised the study.
All authors contributed to the discussion of the results, the manuscript review and improvements.

*Competing interests.*   The authors declare that they have no conflict of interest.

*Acknowledgements.*   This work was partly funded by the German Ministry for Education and Research under grant 01 LG 1907 (project WASCLIM) in the frame of the Role of the Middle Atmosphere in Climate (ROMIC)-program.



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
