# Peer review of "Satellite observations of gravity wave momentum flux in the mesosphere / lower thermosphere (MLT): feasibility and requirements"

_Atmospheric Measurement Techniques, 2022_

## Referee Comment (RC2)

**Review of the manuscript** "Satellite observations of gravity wave momentum flux in the mesosphere / lower thermosphere (MLT): feasibility and requirements" by Chen et al. submitted for a possible publication in AMT [Manuscript ID: #amt-2022-224]

**General comment**

The manuscript investigates requirements for satellite limb optical measurements using $O_2$ A-band emissions to retrieve characteristics of gravity waves (GWs) and GW momentum flux that strongly influences global circulation in the middle and upper atmosphere. The manuscript is mainly based on modelling results. The present study is very useful and worth publication. However, the text is rather demanding to read, partly due to the complexity of the problem. Nevertheless, I believe that some formulations could be simplified, some points better explained and specified or located in more convenient places in the text. I provide several examples in the specific comments below, but I encourage the authors not to limit themselves to them only. I recommend a moderate revision.

**Specific comments**:

-Introduction (for example in Key Quantities), the authors only speak about zonal GW momentum flux and direction distribution of the flux. Does the direction only mean the sign of zonal flux, or also the meridional component. Please explain and reformulate. Why is the meridional component not mentioned in the Introduction section when it is shown in some Figures of the following Sections?

-It is difficult to understand, namely in the Introduction, why "by separately inverting left-hand and right-hand part of the interferogram", independent observation tracks are obtained. Please reformulate or explain better here.

-line 190, *u', v', w'*, define the coordinate system.

Section 2.2. Last sentence. It is partly explained in the Discussion, but here, the meaning of this sentence is quite unclear. Please reformulate/explain or remove.

-line 210, S3D, it should be defined here at the first usage.

-Section 3.1, around line 241, "…*moist convection…*" The moist convection at such high altitudes deserves some explanation.

-Section 3.3. A comparison of usable height ranges for day- and night-time observation should be discussed in more detail.
In addition, HAMMONIA model should be briefly introduced and/or referenced.

-Section 3.5. It should be better explained how two independent temperatures are obtained along the horizontal axis using $O_2$ A-band emissions only.

-last line on page 15, "…*retrieved temperatures, which are about 17 km apart*…". That doesn't make sense to me. Please reformulate.

-Section 3.6. Specify the time interval over which the snapshots used for the tomography are taken. Discuss this time interval with respect to the GW period/wavelength and propagation velocity. Discuss also the assumed angle difference between different positions marked by different colors in Figure 8.

-Section 3.6 or 3.7 (Table 2). Note that the definition of spectral wavenumbers (in cm$^{-1}$) is $1/\lambda$ here, where $\lambda$ is the wavelength, and not $2\pi/\lambda$ which is often used.

-line 393, define FWHM

-Section 3.7, last but one paragraph. The text is difficult to read. Please reformulate/simplify.

-Section 4.2.1, second paragraph "*From the model set-up we expect shortest horizontal wavelengths of O(200km)...*" It should be discussed here that a number of radio and optical observations show shorter wavelengths than 200 km (Nishioka et al., 2013; Chum et al., 2021; Shiokawa et al., 2009; among others)
The authors partly discuss this wavelength limit in the Discussion section and in Appendix E, but this information should be briefly given already here. Moreover, the Discussion section mainly relies on modelling. The already available observations should also be mentioned.

Chum, J., Podolská, K., Rusz, J., Baše, J., Tedoradze, N. (2021), Statistical investigation of gravity wave characteristics in the ionosphere. *Earth Planets Space* **73,** 60, https://doi.org/10.1186/s40623-021-01379-3

Nishioka M, Tsugawa T, Kubota M, Ishii M (2013) Concentric waves and short-period oscillations observed in the ionosphere after the 2013 Moore EF5 tornado. *Geophys Res Lett.* https://doi.org/10.1002/2013GL0579 63

Shiokawa K, Otsuka Y, Ogawa T (2009) Propagation characteristics of nighttime mesospheric and thermospheric waves observed by optical mesosphere thermosphere imagers at middle and low latitudes. *Earth Planets Space 61:479–491.* https ://doi.org/10.1186/BF033 53165

- Figure 13. Specify the time interval (season) for which the Figure was constructed.

-line 622, "*tides cause changes of the large scale winds at similar time scales as the periods of the GWs propagating through these winds*". Specify the periods of tides and GWs considered here.

---

## Author Comment (AC1)

**Response to the referee comments**

2

1

We thank the referees for their valuable and helpful comments. We have addressed all of them one-by-one in details as listed below. The comments are bold and our replies are in regular font. The page/line numbers indicated in our replies are given with respect to the old manuscript, and may differ from the revised manuscript.

**8 Referee #1**

**9 General comments:**

The manuscript is generally well written and well structured. All 10 arguments are clearly described and reasonable, and the conclu-11 sions are justified. The authors present a very comprehensive 12 study, including spectroscopy, instrument design, modeling, and 13 geophysics. My main recommendation, therefore, is that the au-14 thors should keep the focus of the manuscript clearer and reduce 15 side-topics or well-known aspects. That would help the reader to 16 keep oversight over the 19 figures and related descriptions. I rec-17 ommend a minor revision of the manuscript and provide in the 18 following more detailed comments. 19

The referee's comments helped us to improve our manuscript. We went through the manuscript, reduced the side-topics and also highlighted the main questions of this study at the beginning of each of the sub-sections in the Assessment section, in order to keep the readers focused on the key topics. We excluded Appendix A since the discussed scale-separation method was actually not used in our study. We also modified/shortened the Introduction part.

**27 Minor comments:**

1. The Introduction includes an extensive description of the relevance of gravity waves for the understanding of the middle atmosphere. I think this is without doubt, and the description can be
shortened. The description of the MATS mission is from my point
of view not relevant for a feasibility study of another instrument

Following the referee's suggestion, we removed the description about the
MATS mission and rewrote part of the Introduction.

35 2. l. 184/185, Appendix A: As far as I understand, the described
36 method of Ern et al. is not used in this manuscript and the compar37 ison of methods is beyond its scope. Therefore, I suggest removing
38 Appendix A and rephrasing this sentence.

Following the referee's suggestion, we excluded Appendix A describing the
scale separation method, and provided a reference to the paper of Ern et al.
(2018) in former 1.170.

42 3. Fig. 2, Section 2.3: Fig. 2 is very deductive and important for
43 the understanding of the method. I suggest referring the "first
44 question/second question/third question" to the respective up45 per/middle/lower yellow diamonds and adding references to Sec46 tions 4.2 and 4.3.

We thank the referee for this comment, which makes the topics of this
manuscript more focused. We added "Question 1/Question 2/Question 3"
to the three yellow diamond boxes in Fig. 2, and cross-referred to them back
at the beginning of Sections 4.1, 4.2 and 4.3.

**4. l. 505: I do not understand why another cutoff is applied to the simulations compared to the reference. Please describe.**

The cut-off eliminates non-reliable fits where wavelengths are much longer than the analysis volume. In principle, it therefore depends on the number of tracks specifying one of the horizontal cube dimensions. In order not to confuse the reader we have now cut all distributions at 2100 km following the reviewers suggestion. Fig. 10 a-c were updated accordingly.

 $_{58}$  We modified the text in former 1.486-487 as:

59 "All spectra are cut off at longer wavelength of around 2100 km horizontally 60 and 45 km vertically, as the detection upper limit. It results from the limits 61 when filtering reliable fits, which are up to  $\sim 3$  times the cube size, both for 62 horizontal and vertical wavelength." 5. l. 507: It does not become clear to me whether in some cases
(Fig. 10 c and f) there are no waves below 150 km wavelength,
or whether this part of the spectrum is not shown for technical
reasons.

For the 15- and 5-track cases (on sampled data), the wavelength spectra 67 have a clear cutoff of horizontal wavelength close to 150 km at the short 68 end, while the 4- and 2- cases (on retrieved data) have a wider spread of 69 wave distribution towards 100 km, particularly at 75 km. This difference 70 comes from: 1. our simulation is based on the HIAMCM model data, in 71 which the shortest horizontal wavelength that could be resolved is around 72 156 km according to Becker et al. (2022). Therefore, a shortest wavelength 73 of around 150-160 km would be expected in our wave fitting results, and 74 the 15- and 5-track simulation results do conform to this limit by a sharp 75 decrease of spectral power. 2. the general cut-off at the short wavelength 76 side is due to the implementation of the nested interval method in S3D. 77 After a first guess which is in the low frequency region of the spectrum the 78 search region for the minimum depends on the spectral resolution provided 79 by the "natural" spectral grid which a Fourier transform would use. The 80 more points are used (i.e. 2, 5, 15 pts for as many tracks) the narrower is 81 this search region around the initial guess. Accordingly, wave solutions are 82 confined closer to the initial guess and hence in the long wavelength range. 83 Vice versa, for smaller cubes the search region is wider and gets closer to the 84 Nyquist limit - never reaching this, though. This is consistent with the fact 85 that the short-wavelength "cuts" are at wavelength notably longer than the 86 Nyquist wavelength of 60 km. 87

**6. Section 4.3: I recommend referring at the beginning of this section back to Fig. 2 and Section 2.1.**

90 We added reference to Fig. 2 and Section 2. at the beginning of Section 4.3.

7. Section 5 discusses in general the relevance of the examination
of mesoscale gravity waves independent from the proposed instrument or the availability of additional information (wave sources,
winds, ...). I suggest either shortening this section or pointing
out why these studies cannot be done with other (existing) instruments.

In Sect. 5 after former 1.609 we now have added a short summary of existing
satellite observations of GWs in the MLT region. None of these data sets
provides 3D information of the observed GWs, which strongly limits the
interpretation of these data.

8. l. 610 – l. 613: I agree that wind information is crucial for the
understanding of wave dissipation and other processes. However,
it is hardly available on a global scale in the MLT. Wind data from
assimilated temperature information may lack precision, especially
for non-linear processes. Effects like GW bending cannot be acknowledged at all. Please comment on the consequences of limited
data availability for the science questions.

Unlike in the stratosphere, mesospheric assimilation systems have far too low 108 resolution to resolve GWs in any realistic fashion. This is, however, not re-109 quired for the interpretation of the data. We need only a reasonably realistic 110 representation of the global wind fields. This naturally contains gradients, 111 both in the vertical and in the horizontal and hence will cause ray-tracing 112 modelling to produce refraction of the wave vector (both horizontal and ver-113 tical components) and to generate critical levels when  $\lambda_z \to 0$ . The question 114 then is not whether the fields are sufficiently accurate, but sufficiently re-115 alistic. Assimilation systems still struggle, in particular in times of special 116 interest (Harvey et al., 2022), as apparently the information content of the 117 observations has insufficient weight to correct the model. Still, one can use 118 geostrophic winds as an approximation of the large scale flow. In addition, 119 methods were developed to determine the tides (Nguyen and Palo, 2013; Pe-120 datella et al., 2016), and based on such results one could adapt tidal models 121 such as the GSWM in order to gain a complete view of all variables. The 122 current focus on the MLT rises therefore hope that methods will be found 123 to determine sufficiently realistic winds to produce diagrams of the ground-124 based phase speed, assess critical level filtering and perform ray-tracing to 125 investigate the fate and characterize the origin of GWs in a general fashion. 126 Whether we will gain sufficiently accurate winds for backward ray-tracing of 127 such mesoscale GWs by 60 to 80 km altitude towards individual sources is 128 a different matter and there I am less optimistic. The need for only global 129 fields is now emphasized and a shorter version of the state of assimilation 130 and other methods to determine is included into the text. 131

Of course, it is not sure whether global wind observations, or reliable winds from data assimilation, will be available in the upper mesosphere/lower thermosphere at the time the instrument will be in operation. This means, as already stated in former lines 636–638, it is not sure whether we will be able to perform reliable ray tracing of gravity waves, and identification of the gravity wave sources. Also gravity dissipation studies in relation to the background wind would not be possible.

139 However, the gravity wave data set that we expect to obtain from this novel

observation method will be quite unique, and of great value in itself. Even 140 without wind observations, studies based on the observed directional grav-141 ity wave momentum flux can be performed in a climatological sense, for 142 example by comparison with zonal wind climatologies, or climatologies of 143 atmospheric tides. Particularly, the interaction between gravity waves and 144 tides is not well understood and offers a wide field of applications. Further, 145 the novel gravity wave data set can be used to identify cases of excitation of 146 secondary gravity waves. This can be performed by identifying fishbone-like 147 structures in along-track/altitude cross sections without the need of having 148 background wind information. For these kind of studies the relatively short 149 along-track sampling of 30km, combined with a tomographic retrieval, will 150 be very beneficial. 151

This reasoning will be added in the revised manuscript after former 1.618 & 1.640.

**9. l. 643/644: I suggest comparing to other global observations instead of comparing to models. Even GW-resolving GCMs may not display true atmospheric states despite they are good tools for the understanding of atmospheric processes.**

At the end, our point is that we need good quantitative global measures 158 of GWMF. We don't have these, yet. Our current error bar on the global 159 climatologies which we can deduce e.g. from SABER is about a factor of 160 3 and with that you can argue for everything between these scales being a 161 minor contribution and the only thing worthwhile looking at. My best guess 162 would be about half from scales longer than 100km and half from scales 163 shorter, but nobody knows. We have included the reference to the shorter 164 scales here to indicate the contradicting evidence. We have tried to sharpen 165 the point in the text by including a cross-reference to the introduction, the 166 short-scale observations you mentioned and making evident that all this is 167 partly contradictory evidence. 168

A summary of existing satellite observations in the MLT region, and of their limitations has been added in Sect. 5.

**171 10. l. 653: I suggest describing the effects of the observational filter 172 much earlier.**

173 We added the sentence and reference about the observational filter in the 174 Introduction after former 1.110.

175 11. l. 676/677: I agree with the statement about zonal mean cli-176 matologies but suggest removing the two lines including wind data.

**177 The authors describe their own concerns in L. 636-638.**

We removed the sentence in former 1.676/677 in the conclusions.

**179 Technical comments and typos:**

- 180 1. l. 63: "residual"
- 181 We changed the text accordingly.
- 182 2. l. 429: "shown by Lehmann et al. (2012)"
- 183 We changed the text accordingly.
- 184 3. l. 439: "(cf. Section 2.1 and Figure 2)"
- 185 We added in the text accordingly.
- 186 4. l. 450: "by Ern et al. (2004)"
- 187 We changed the text accordingly.
- 188 5. l. 470: I suggest adding "(see data flow to the middle yellow 189 diamond in Fig. 2)"
- 190 We added in the text accordingly.
- 191 6. l. 498: "5-track"
- 192 We changed the text accordingly.
- 193 7. l. 583: "can be generated" should read "can be calculated"
- 194 We changed the text accordingly.

**References**

- Becker, E., Vadas, S. L., Bossert, K., Harvey, V. L., Zülicke, C. and
  Hoffmann, L. (2022), 'A high-resolution whole-atmosphere model with resolved gravity waves and specified large-scale dynamics in the troposphere
  and stratosphere', J. Geophys. Res. Atmos. 127(2), e2021JD035018.
  e2021JD035018 2021JD035018.
- URL: https://aqupubs.onlinelibrary.wiley.com/doi/abs/10.1029/2021JD035018

- 202 Ern, M., Trinh, Q. T., Preusse, P., Gille, J. C., Mlynczak, M. G., Russell III,
- J. M. and Riese, M. (2018), 'GRACILE: A comprehensive climatology of
- atmospheric gravity wave parameters based on satellite limb soundings',
- 205 Earth Syst. Sci. Dat. **10**, 857–892.
- 206 URL: https://www.earth-syst-sci-data.net/10/857/2018/
- Harvey, V. L., Pedatella, N., Becker, E. and Randall, C. (2022), 'Evaluation
  of polar winter mesopause wind in WACCMX+DART', J. Geophys. Res.
  Atmos. 127(15), e2022JD037063.
- 210 URL: https://agupubs.onlinelibrary.wiley.com/doi/abs/10.1029/2022JD037063
- Nguyen, V. and Palo, S. E. (2013), 'Technique to produce daily estimates of
  the migrating diurnal tide using TIMED/SABER and EOS Aura/MLS',
  J. Atm. Sol.-Terr. Phys. 105, 39–53.
- Pedatella, N. M., Oberheide, J., Sutton, E. K., Liu, H. L., Anderson,
  J. L. and Raeder, K. (2016), 'Short-term nonmigrating tide variability
  in the mesosphere, thermosphere, and ionosphere', *J. Geophys. Res. Space*121(4), 3621–3633.

---

## Author Comment (AC2)

**Response to the referee comments**

We thank the referees for their valuable and helpful comments. We have addressed all of them one-by-one in details as listed below. The comments are in bold and our replies are in regular font. The page/line numbers indicated in our replies are given with respect to the old manuscript, and may differ from the revised manuscript.

**Referee #2**

**General comments:**

**The manuscript investigates requirements for satellite limb optical measurements using $O_2$ A-band emissions to retrieve characteristics of gravity waves (GWs) and GW momentum flux that strongly influences global circulation in the middle and upper atmosphere. The manuscript is mainly based on modelling results. The present study is very useful and worth publication. However, the text is rather demanding to read, partly due to the complexity of the problem. Nevertheless, I believe that some formulations could be simplified, some points better explained and specified or located in more convenient places in the text. I provide several examples in the specific comments below, but I encourage the authors not to limit themselves to them only. I recommend a moderate revision.**

The referee's comments helped us to improve our manuscript. We revised the manuscript thoroughly, reduced/simplified the side-topics and rephrased some parts of the text to make them more clear. Specifically, we excluded Appendix A since the discussed scale-separation method was actually not used in our study. The explanation about the Key Quantity – zonal mean GW momentum flux was updated in the Introduction and Sect. 2.2. The description of the interferogram split method in Sect. 3.5 was reformulated. In the Discussion section, we included the discussions about the existing satellite observations and their limitations, as well as the global wind data availability in the MLT region.

**Specific comments**

**1. Introduction (for example in Key Quantities), the authors only speak about zonal GW momentum flux and direction distribution of the flux. Does the direction only mean the sign of zonal flux, or also the meridional component. Please explain and reformulate. Why is the meridional component not mentioned in the Introduction section when it is shown in some Figures of the following Sections?**

For the Key Quantities we considered the zonal mean GW momentum flux, i.e., the zonally averaged vertical flux of horizontal momentum of GWs, since it can be directly inferred from the wind data and can thus serve as an absolute reference for global GW characterization, described in more detail in Section 2.2. The direction refers to the sign of the zonal mean GW momentum flux, which itself consists of two components: zonal component $F_{px}$ and meridional component $F_{py}$. The two, i.e., zonal and meridional, components of zonal mean GW momentum flux are illustrated in left and right panels respectively in Fig.9 and Fig.13-16.

Regarding to the referee's comment, we added the detailed explanation in former l.94-96 in the Introduction:

"In order to close the momentum budget, in particular the zonal mean of the zonal GW momentum flux is required, but zonal mean meridional momentum flux may contribute as well (Ern et al., 2013). ...

For our study the zonal mean of zonal GW momentum flux is of particular importance as the values directly inferred from the winds provide a true reference value. This is, to a somewhat lesser degree, also true for the meridional momentum flux, as will be discussed below."

**2. It is difficult to understand, namely in the Introduction, why "by separately inverting left-hand and right-hand part of the interferogram", independent observation tracks are obtained. Please reformulate or explain better here.**

For a better understanding, we reformulated this sentence as "by splitting one interferogram into two left-hand and right-hand parts and separately mirroring each parts (cf. Sect. 3.5), " in former l.137 in the Introduction and referred to the corresponding Section 3.5 for a detailed method description.

**3. line 190, $u', v', w'$, define the coordinate system.**

As recommended, we added the definition of the coordinate system after former l.190.

**4. Section 2.2. Last sentence. It is partly explained in the Discussion, but here, the meaning of this sentence is quite unclear. Please reformulate/explain or remove.**

Following the referee's suggestion, we removed the last sentence from Sect. 2.2.

**5. line 210, S3D, it should be defined here at the first usage.**

As recommended, we added the definition of S3D after former l.210.

**6. Section 3.1, around line 241, "…moist convection…" The moist convection at such high altitudes deserves some explanation.**

Though of course there is no moist convection at the observation altitude it is one of the important sources of the waves which govern this height region: GWs, tides and equatorial wave modes. This explanation was included in the revised text after former l.241.

**7. Section 3.3. A comparison of usable height ranges for day- and night-time observation should be discussed in more detail.**

We considered for the daytime an observation altitude region of 60-120 km, which was reduced to the range of 80 km to 100 km during nighttime as only the photochemical production channel exists.

We added the corresponding description about the altitude range at the end of Section 3.3.

**In addition, HAMMONIA model should be briefly introduced and/or referenced.**

We added the reference to the HAMMONIA model data in former l.283.

**8. Section 3.5. It should be better explained how two independent temperatures are obtained along the horizontal axis using $O_2$ A-band emissions only.**

For clarification, we reformulated most of the description about the interferogram split method in Section 3.5.

**9. last line on page 15, "…retrieved temperatures, which are about 17 km apart…". That doesn't make sense to me. Please reformulate.**

This comment is related to the previous one. We have reformulated most of Sect. 3.5, which should make this point much clearer.

**10. Section 3.6. Specify the time interval over which the snapshots used for the tomography are taken. Discuss this time interval with respect to the GW period/wavelength and propagation velocity. Discuss also the assumed angle difference between different positions marked by different colors in Figure 8.**

Looking at the individual "rays" of measurements from the simulations, one can analyse, where an overlap occurs. For the given geometry, overlaps occur for measurements up to 160s apart. This largest time difference for this backwards-looking instrument occurs between measurements at high angles, i.e. tangent point altitude at 120km, and later measurements at low angles, i.e. tangent point altitudes at 70km.

As most information is gained from the emissions around the tangent point, the practical time delta is more in the order of 80s.

This is at least one order of magnitude smaller than the periods of GWs that our proposed instrument is sensitive to: We aim at GWs of $\lambda_h > 100$ km and $\lambda_z \approx 10$ km which corresponds to an intrinsic period of roughly one hour. By Doppler shift shorter ground based periods may occur, but it is expected that the bulk of the observed GWs has ground-based periods of a few hours.

The angular differences are rather small for a tomographic method and form an extreme case of limited-angle-tomography. For the proposed retrieval scheme, the different overlaps of line-of-sights as well as the exponential increase of number densities to lower altitudes are more important for localisation of information.

We added the following sentence to the main text in former l.385:

"The satellite speed allows to gather all relevant measurements for on spatial sample in the order of minutes, which is short compared to typical periods of gravity waves observable by our instrument."

**11. Section 3.6 or 3.7 (Table 2). Note that the definition of spectral wavenumbers (in cm$^{-1}$)is $1/\lambda$ here, where $\lambda$ is the wavelength, and not $2\pi/\lambda$ which is often used.**

We added a footnote in Table 2 for to remind of the definition of spectral wavenumber.

**12. line 393, define FWHM**

We added the definition of FWHM in former l.393.

**13. Section 3.7, last but one paragraph. The text is difficult to read. Please reformulate/simplify.**

Following the referee's suggestion, we rephrased this paragraph as below:

"The synthetic observation data have a fixed sampling in $x$, $y$ and $z$ direction, on which the analysis cube size is defined via the number of sampling points. For the model data, a fixed model sampling in terms of degrees longitude in zonal direction means a coarser (in distance) sampling close to the equator and a finer sampling at high latitudes due to the shorter distance between two respective longitudes at higher latitudes. Therefore, the size of a fixed cube is specified in kilometers instead of degrees and the number of fitting points is adapted accordingly. This ensures that the same part of the spectrum is targeted independent of latitude along the longitude direction."

**14. Section 4.2.1, second paragraph " From the model set-up we expect shortest horizontal wavelengths of O(200km) . . . " It should be discussed here that a number of radio and optical observations show shorter wavelengths than 200 km (Nishioka et al., 2013; Chum et al., 2021; Shiokawa et al., 2009; among others).**

For various reasons we would have preferred, of course, a model with higher resolution encompassing the entire MLT. At the end we have to take what is feasible nowadays. The fact that short waves must not be neglected, has been now included in Sect. 4.2.1 after former l.477 and also in the discussion after former l.643.

**The authors partly discuss this wavelength limit in the Discussion section and in Appendix E, but this information should be briefly given already here. Moreover, the Discussion section mainly relies on modelling. The already available observations should also be mentioned.**

**Chum, J., Podolská, K., Rusz, J., Baše, J., Tedoradze, N. (2021), Statistical investigation of gravity wave characteristics in the ionosphere. Earth Planets Space 73, 60, https://doi.org/10.1186/s40623-021-01379-3**

**Nishioka M, Tsugawa T, Kubota M, Ishii M (2013) Concentric waves and short-period oscillations observed in the ionosphere after the 2013 Moore EF5 tornado. Geophys Res Lett. https://doi.org/10.1002/2013GL0579 63**

**Shiokawa K, Otsuka Y, Ogawa T (2009) Propagation characteristics of nighttime mesospheric and thermospheric waves observed by optical mesosphere thermosphere imagers at middle and low latitudes. Earth Planets Space 61:479–491. https://doi.org/10.1186/BF033 53165**

True. we have included also a reference to both short and mesoscale wavelength observations in the discussion. This hopefully clarifies that we need to have new observations in order to identify the relative contribution of different scales.

15. **Figure 13. Specify the time interval (season) for which the Figure was constructed.**

We added in the caption "01-Jan-2016 06 UT" and in the text in former l.541: "for 01-Jan-2016 06 UT (i.e., winter in the northern hemisphere and summer in the southern hemisphere)" to specify the season in Fig. 13.

16. **line 622, " tides cause changes of the large scale winds at similar time scales as the periods of the GWs propagating through these winds ". Specify the periods of tides and GWs considered here.**

We added "e.g., diurnal and semi-diurnal tides," in former l.622 to specify the periods.

**References**

Ern, M., Arras, C., Faber, A., Fröhlich, K., Jacobi, C., Kalisch, S., Krebsbach, M., Preusse, P., Schmidt, T. and Wickert, J. (2013), Vertical coupling by gravity waves in atmospheric dynamics: Observations, ray tracing, and implications for global modeling, *in* Franz-Josef Lübken, ed., 'Climate and Waether of the Sun-Earth System (CAWSES)', Springer Atmospheric Sciences, Dordrecht, Netherlands, pp. 383–408.